# SimpleTM: A Simple Baseline for Multivariate Time Series Forecasting

**Hui Chen**[1]    **Viet Luong**[1]    **Lopamudra Mukherjee**[2]    **Vikas Singh**[1] *
[1]University of Wisconsin-Madison    [2]University of Wisconsin-Whitewater
`hchen795@wisc.edu`   `vhluong@wisc.edu`
`mukherjl@uww.edu`   `vsingh@biostat.wisc.edu`

## Abstract

The versatility of large Transformer-based models has led to many efforts focused on adaptations to other modalities, including time-series data. For instance, one could start from a pre-trained checkpoint of a large language model and attach adapters to recast the new modality (e.g., time-series) as "language". Alternatively, one can use a suitably large Transformer-based model, and make some modifications for time-series data. These ideas offer good performance across available benchmarks. But temporal data are quite heterogeneous (e.g., wearable sensors, physiological measurements in healthcare), and unlike text/image corpus, much of it is not publicly available. So, these models need a fair bit of domain-specific fine-tuning to achieve good performance – this is often expensive or difficult with limited resources. In this paper, we study and characterize the performance profile of a non-generalist approach: our SimpleTM model is specialized for multivariate time-series forecasting. By simple, we mean that the model is lightweight. It is restricted to tokenization based on textbook signal processing ideas (shown to be effective in vision) which are then allowed to attend/interact: via self-attention but also via ways that are a bit more general than dot-product attention, accomplished via basic geometric algebra operations. We show that even a single- or two-layer model gives results that are competitive with much bigger models, including large transformer-based architectures, on most benchmarks commonly reported in the literature.

## 1 Introduction

Multivariate time-series (MTS) data are ubiquitous in various disciplines such as finance and economics Andersen et al. (2006), climate science Mudelsee (2019), healthcare Zeger et al. (2006), geophysics Gubbins (2004), and industrial monitoring Truong et al. (2022). Consequently, MTS data processing and analysis techniques have been extensively studied, going back to works in vector autoregressive models Lütkepohl (2013), dynamic factor models Molenaar et al. (1992), state-space models Rangapuram et al. (2018) and others. The literature provides rich theory and various solutions depending on the assumptions that make the most sense for the data at hand, e.g., homoscedasticity versus heteroscedasticity Rodríguez & Ruiz (2005), degree of autocorrelation Bence (1995), and stationarity versus non-stationarity Das & Nason (2016). Such models refined over decades are well-studied in economics, computational finance and statistics. While progress in deep learning architectures over the last 10+ years has led to the most significant gains in performance capabilities for tasks involving image and natural language data, there is a growing body of literature (discussed below) describing strategies for harnessing these models for multivariate time-series data Liu et al. (2024a); Huang et al. (2023); Zhang & Yan (2023).

**Deep Architectures for MTS data.** Most types of widely used deep architectures – from convolutional neural networks LeCun et al. (1998); Simonyan & Zisserman (2015); He et al. (2016) to graph neural networks (GNN) Kipf & Welling (2017); Hamilton et al. (2017) to transformers Vaswani et al. (2017); Devlin et al. (2019); Dong et al. (2021) – have all been adapted and attempted for various types of MTS data Bagnall et al. (2018). For instance, Zhang & Yan (2023); Zhou et al.

---

*Code is available at GitHub: `https://github.com/vsingh-group/SimpleTM`.

(2022b); Wu et al. (2021); Liu et al. (2024a) use an attention mechanism to model the long-term interaction between different time points whereas approaches using GNNs Cheng et al. (2022); Li et al. (2023); Jin et al. (2023) seek to extract interaction adaptively between different time-series. However, as noted in Huang et al. (2023), all methods face challenges in handling temporal fluctuations and heterogeneity between variables (i.e., different time-series in the same data). But perhaps more importantly, there is an immense degree of variability between different MTS datasets. For instance, MTS data from wearable sensors will bear little to no similarity to electroencephalogram (EEG) recordings of brain activity. We know that internet-scale text and image data corpus have been used, to train large language and vision models, where the sheer size of the dataset provides the model some ability to handle heterogeneity. But while the raw sizes of MTS data produced or acquired each day (e.g., in physiological recordings or wearable sensors) is enormous, only a minor fraction of it is publicly available due to strict privacy regulations (HIPAA) or laws surrounding sharing of consumer behavior data or entirely non-legal reasons (proprietary, competitiveness). To summarize, such data remain scarce and thereby, deploying pre-trained models in a specific setting involving our own MTS dataset requires care and often significant fine-tuning. In fact, Zeng et al. (2023) found that for a number of publicly available datasets, a simple one-layer linear model can frequently outperform generic approaches based on Transformers, suggesting that translating the same backbone to complex and heterogeneous MTS data is challenging. Promisingly, in the last year, a number of interesting approaches Liu et al. (2024a); Wang et al. (2024); Nie et al. (2023); Chen et al. (2024), have been proposed which make specific adjustments/modifications to the architecture to better handle the nuances and complexity of MTS data, and show robust/reproducible performance. Several of these models will serve as our baselines later.

**Repurposing LLMs for Time-series data.** A related but distinct line of work seeks to re-interpret time-series data as natural language, and allows operating on top of powerful large language models Jin et al. (2024). Such an approach can benefit from the vast amount of text data the language model has already been trained on, which is kept frozen, and one assumes that a mechanism to map chunks of time-series to word embeddings can be estimated based on a sufficiently large MTS dataset. This mapping is often accomplished by training specialized adapters placed before and after the LLM in the pipeline. This direction is evolving rapidly and providing promising results, but as of now, deploying the model on a domain specific dataset with its own specific characteristics of stationarity and seasonality, while possible, remains quite compute intensive.

**This work.** Our paper aligns more closely with the aforementioned *non-generalist* approaches in that the intended use of the model will only be for multi-variate time-series data. Instead of modifying a large Transformer-based backbone, we will add in modules, one by one, quite conservatively. Similar to LLMs, we also use tokenization but given the well-defined application scope (time-series data), we will use ideas based directly on classical signal processing Haykin & Veen (1998). Then, we borrow the self-attention module and make a small modification to it, so it can capture a richer dependency structure between tokens, allowing it to capture dependencies across-time and across-dimensions. The **key contributions** of our work are summarized as follows:

**(i)** We propose *SimpleTM*, a simple yet effective architecture that uniquely integrates classical signal processing ideas with a slightly modified attention mechanism.

**(ii)** We show that even a single-layer configuration can effectively capture intricate dependencies in multivariate time-series data, while maintaining minimal model complexity and parameter requirements. This streamlined construction achieves a performance profile surpassing (or on par with) most existing baselines across nearly all publicly available benchmarks.

## 2 Preliminaries: Problem setup and Notations

**Univariate time-series.** Let $(x_1, \cdots, x_L)$ be a single historical (or lookback) time-series of length $L$ where $x_t \in \mathbb{R}$ denotes the measurement/value at the $t$-th timestep. Let $(y_1, \cdots, y_H)$ be a single time-series of length $H$ in the future. We call $H$ the forecast/horizon window length and $L$ the lookback window length. Time-series forecasting asks if we can predict $(y_1, \cdots, y_H)$ from $(x_1, \cdots, x_L)$.

**Multi-variate time-series.** Let $\mathbf{X} \in \mathbb{R}^{C \times L}$ and $\mathbf{Y} \in \mathbb{R}^{C \times H}$ be two matrices, jointly drawn from some distribution $\mathcal{P}$. We also write $\mathbf{x}_t$ and $\mathbf{y}_t$ as the $t$-th column of $\mathbf{X}$ and $\mathbf{Y}$ respectively. That is, we observe $L$ measurements for each of $C$ channels or variables from $\mathbf{X}$. Our goal is to "forecast"

the time-series of $H$ timesteps, each timestep $t$ in the forecast window is a vector of length $C$, collectively called $\mathbf{Y}$.

**Multiple Multi-variate time-series.** Denote by $N$ the sample size: the number of multi-variate time-series we observe. We can use $i$ as a generic index for a specific sample for $i \in [N]$.

*Remark.* This multivariate setting captures scenarios where we are measuring time-series data for $C$ different channels or variables in a synchronized manner, which becomes particularly valuable when there are correlations or dependencies among these variables.

**Definition 1 (Forecasting error)** *Assume a multi-variate time-series* $(\mathbf{X}, \mathbf{Y}) \sim \mathcal{P}$, *where* $\mathbf{X} \in \mathbb{R}^{C \times L}$ *and* $\mathbf{Y} \in \mathbb{R}^{C \times H}$. *For any mapping* $f : \mathbb{R}^{C \times L} \to \mathbb{R}^{C \times H}$, *we call it a forecasting function. We define the forecasting error with regards to* $f$ *as*

$$\mathcal{L}(f) := \mathbb{E}_{(\mathbf{X}, \mathbf{Y}) \sim \mathcal{P}} ||\mathbf{Y} - f(\mathbf{X})||_{\mathcal{F}}, \tag{1}$$

*where* $|| \cdot ||_{\mathcal{F}}$ *denotes the Frobeneus norm of the matrix and* $\mathbb{E}$ *is the expectation over the joint distribution* $\mathcal{P}$. *We are given a set of* $i.i.d.$ *samples* $\{(\mathbf{X}_i, \mathbf{Y}_i)\}_{i=1}^{N}$. *We define the empirical risk with regard to forecasting function* $f$ *as*

$$L(f) := \frac{1}{N} \sum_{i=1}^{N} ||\mathbf{Y}^{(i)} - f(\mathbf{X}^{(i)})||_{\mathcal{F}}. \tag{2}$$

Our goal is to optimize over $f$ to minimize the empirical forecasting error. We will now introduce the specific modules in our overall model, after which we will describe the experimental evaluations.

## 3 MODULE 1: TOKENIZATION VIA STATIONARY WAVELET TRANSFORM

**Motivation/Rationale.** For this first module, we seek a tokenization scheme for MTS data that relieves the amount of work that the downstream modules need to do – discovering all local/global dependencies – which raises both the compute footprint and also the sample sizes needed. Ideally, if our tokens could capture temporal information across multiple scales (rapid, short-term variations to slow, long-term trends), and capture both local/global patterns within each of the $C$ variables/sites, then the task of exactly how to synthesize this information for forecasting $\mathbf{Y}$ would be simplified. If we can allow scale-specific processing, then arguably the synthesis task can benefit from the specific modules processing each scale, acting collaboratively.

**One possible solution.** The reader will immediately see that the Wavelet transform is a first principles based solution to the requirements outlined above, and this idea has recently found use in processing

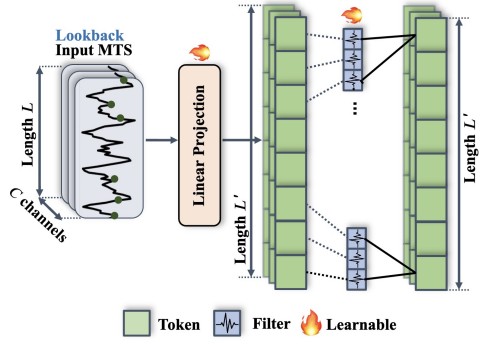

Figure 1: For each channel, time-series measurements are passed through a stationary wavelet transform followed by a linear projection to obtain $L'$ tokens.

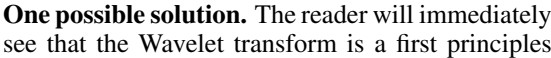

image data in Transformer models Yao et al. (2022); Zhu & Soricut (2024). It offers a multi-scale decomposition of each signal while maintaining temporal localization. We will treat each wavelet scale *separately* which will allow learning scale-specific interactions within each variable. If our forecast window is dominated by dependencies that are prominent at one scale but not the other, such a construction provides the downstream modules relevant information to operate with.

**Details of the construction.** We now present our tokenization scheme. Let $\boldsymbol{X} = \{\boldsymbol{x}_1, \cdots, \boldsymbol{x}_L\} \in \mathbb{R}^{C \times L}$ denote our multivariate time-series, where $C$ is the number of channels as before.

*A) Linear Projection.* We first apply a linear projection $g(\cdot; \boldsymbol{\theta}) : \mathbb{R}^L \to \mathbb{R}^{L'}$ to embed each channel into a hidden/latent space which gives $\tilde{\boldsymbol{X}} = \{\tilde{\boldsymbol{x}}_1, \tilde{\boldsymbol{x}}_2, \cdots, \tilde{\boldsymbol{x}}_{L'}\} = g(\boldsymbol{X})$.

*B) Stationary Wavelet Transform (SWT).* To achieve a multi-scale representation, we use a learnable stationary wavelet transformation (SWT). SWT Nason & Silverman (1995) is defined as

$$SWT(\cdot; \boldsymbol{h}_0, \boldsymbol{g}_0) : \mathbb{R}^{C \times L'} \to \mathbb{R}^{C \times L' \times (S+1)},$$

where $\boldsymbol{h}_0, \boldsymbol{g}_0 \in \mathbb{R}^{C \times k}$ are learnable filters with kernel size $k$, and $S$ is the decomposition level. This transformation produces a set of time-frequency tokens $\{\boldsymbol{u}_1^{(s)}, \boldsymbol{u}_2^{(s)}, \ldots, \boldsymbol{u}_{L'}^{(s)}\}_{s=0}^{S}$, capturing information at different temporal scales for each channel independently, see Figs. 1–2. SWT is suitable since it provides a time-invariant decomposition while preserving the original temporal structure. This is achieved by avoiding downsampling at each decomposition level, thus maintaining the up-scaled length/size $L'$. SWT is also shift-invariant making it effective in capturing localized events across multiple scales. At the core of the SWT are the mother wavelet $\psi(t)$ and scaling function $\phi(t)$. The family of discrete wavelets can be expressed as:

$$\psi_{s,k}(t) = 2^{-s/2}\psi(2^{-s}t - k) \quad \text{and} \quad \phi_{s,k}(t) = 2^{-s/2}\phi(2^{-s}t - k),$$

where $s$ controls the scale (dilation) and $k$ determines the position (translation).

*C) Obtaining Wavelet Coefficients.* The embedded time series $\{\tilde{x}_t\}_{t=1}^{L'}$ undergoes decomposition via the stationary wavelet transform (SWT), yielding approximation coefficients $a_t^{(s)}$ and detail coefficients $u_t^{(s)}$ at each level $s$. For clarity, we present the process for a univariate series. SWT uses two main filters: a low-pass filter $h$ and a high-pass filter $g$, derived from the scaling function $\phi(t)$ and the wavelet function $\psi(t)$ respectively:

$$h(k) = \langle \phi(t), \phi(2t - k) \rangle \quad \text{and} \quad g(k) = \langle \psi(t), \phi(2t - k) \rangle.$$

Starting with $a_t^{(0)} = \tilde{x}_t$, the decomposition at level $s$ is computed as:

$$a_t^{(s+1)} = \sum_k h^{(s)}(k)a_{t+k}^{(s)} \quad \text{and} \quad u_t^{(s+1)} = \sum_k g^{(s)}(k)a_{t+k}^{(s)}.$$

Here, $h^{(s)}$ and $g^{(s)}$ are upsampled versions of $h$ and $g$, obtained by inserting $2^s - 1$ zeros between each original

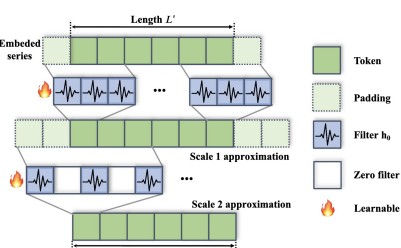

Figure 2: Tokenization via SWT: The input series is padded and processed through learnable filters. At each scale, SWT inserts zeros between filter coefficients, producing a non-decimated output. This approach, demonstrated for a scale-2 approximation, allows shift-invariant feature extraction while preserving temporal resolution.

filter coefficient. This upsampling preserves the signal length, ensuring time invariance. Instead of keeping filter coefficients fixed, we allow the coefficients $h$ and $g$ to adapt to the data, i.e., making them learnable (Michau et al., 2022) allowing them to capture relevant patterns and features at each variate level more effectively. Our experiments demonstrate that the learned filters exhibit correlation patterns that resemble those in the respective variables/channels but switching this adaptivity/learning off does not adversely impact the results much.

**Summary of tokenization scheme.** The iterative decomposition yields a final approximation $u_t^{(0)} = a_t^{(S)}$ and a set of wavelet coefficients $\{u_t^{(s)}\}_{s=1}^{S}$ at each time point $t$ across various scales. This decomposition allows for a complete reconstruction of the up-projected time series if desired:

$$\tilde{x}_t = \sum k u_k^{(0)} \phi_{S,k}^{(t)} + \sum_{s=1}^{S} \sum_k u_k^{(s)} \psi_{s,k}(t). \tag{3}$$

In our tokenization scheme, each time-frequency point $u_t^{(s)}$ serves as a **token**, encapsulating information at a specific scale $s$ and time $t$. This multi-resolution representation provides a rich, structured view of the data, where each token inherently retains both its temporal context and frequency information. This approach is simple but aligns well with our initial objectives.

## 4 MODULE 2: A SMALL MODIFICATION OF SELF-ATTENTION

**Motivation.** Recall that each token represents multiple channels at a specific "pseudo" time point (pseudo because the length is $L'$ and not $L$) for a specific wavelet scale. SWT already captures some temporal/frequency information. But we also want to characterize the full range of inter-channel dynamics, cheaply. In finance, some asset prices move together and others move inversely, and this can change over time. Tokens from a fine resolution might show high linear independence for a

rapidly changing variable, capturing short-term dynamics, while those from coarser scales can reveal long-term correlations between different channel subsets, reflecting slower, persistent patterns. The degree of inter-channel complementarity or linear independence is not fully encoded by a scalar. In a five channel (or variable) system, tokens $(1, 1, 0, 0, 0)$ and $(0, 0, 1, 1, 0)$ give a zero dot product, but span a 4D subspace, indicating high complementarity. This could reveal, for instance, that the first two channels and the next two channels are behaving as coupled pairs. Leveraging such information explicitly may be unnecessary in a large Transformer model with many layers – where we conjecture that these complex dependencies may get picked up anyway. But in a smaller model, endowing the model with such a capability explicitly appears like a good idea.

**One possible solution.** It turns out that geometric algebra product Artin (2011) offers these abilities. It extends classical linear algebra to provide a unified setup for expressing geometric constructions. Put simply, we obtain a small generalization of self-attention which preserves the capabilities of standard dot-product attention. Note that Transformer models based on Clifford Algebra have been proposed recently Brehmer et al. (2023); de Haan et al. (2024) – these are broadly applicable but computationally heavy, see Chytas et al. (2024). This is because the size of the geometric product scales exponentially with the number of dimensions involved in the product. Our design is quite light, involves minimal changes to self-attention and well suited for our problem.

**Details of the construction.** We summarize a few concepts before describing the low-level details.

*A) Brief Geometric Algebra Review.* Geometric Algebra (GA) provides a framework for representing and manipulating geometric objects. We focus on $G_2$, the GA over a 2-dimensional vector space because we consider *pairs* of tokens in our attention mechanism, regardless of the tokens' dimensionality. The fundamental object in $G_2$ is the multivector, expressed as $M = \langle M \rangle_0 + \langle M \rangle_1 + \langle M \rangle_2$, where $\langle M \rangle_k$ is the $k-$vector part of $M$ for $k \in \{0, 1, 2\}$. The key operation in GA is the geometric product, denoted by: $\boldsymbol{\alpha\beta} = \boldsymbol{\alpha} \cdot \boldsymbol{\beta} + \boldsymbol{\alpha} \wedge \boldsymbol{\beta}$, where $\cdot$

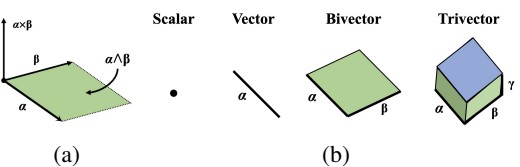

Figure 3: Geometric product objects. (a) shows the oriented parallelogram of the wedge product $\alpha \wedge \beta$ while (b) shows the progression from scalars to vectors, bivectors, and trivectors.

denotes the dot product and $\wedge$ denotes the wedge (or outer) product. The wedge product $\wedge$, also known as the exterior product, represents the oriented area of the parallelogram spanned by two vectors. For vectors $\boldsymbol{\alpha}$ and $\boldsymbol{\beta}$, the wedge product $\boldsymbol{\alpha} \wedge \boldsymbol{\beta}$ results in a bivector (a 2-dimensional element in the algebra). As an example in $G_2$, consider $\boldsymbol{\alpha} = a\mathbf{e}_1 + b\mathbf{e}_2$ and $\boldsymbol{\beta} = c\mathbf{e}_1 + d\mathbf{e}_2$, where $\mathbf{e}_1$ and $\mathbf{e}_2$ are orthonormal basis vectors. Their wedge product is $\boldsymbol{\alpha} \wedge \boldsymbol{\beta} = (ad - bc)(\mathbf{e}_1 \wedge \mathbf{e}_2)$. Here $(ad - bc)$ represents the area magnitude, while $\mathbf{e}_1 \wedge \mathbf{e}_2$ indicates the orientation in the plane.

*B) Instantiating Geometric Product in our case.* We can reformulate the attention mechanism using the geometric product. For tokens $t$ and $t'$, instead of just computing their dot product, we can use the geometric product which combines the dot product (scalar part) with the wedge product (bivector part), encoding both magnitude-based similarity and geometric relationships between the tokens. So, we capture not only the scalar similarity between tokens (via dot product) but also their linear independence and the orientation of the space they span (via wedge product). This allows detecting complementary information across channels and changing inter-channel dynamics. For two tokens $\boldsymbol{\alpha}$ and $\boldsymbol{\beta}$ for different time points across $C$ channels, the $\boldsymbol{\alpha} \cdot \boldsymbol{\beta}$ part is the scalar similarity, while $\boldsymbol{\alpha} \wedge \boldsymbol{\beta}$ tells us how these tokens complement each other across the $C$ channels.

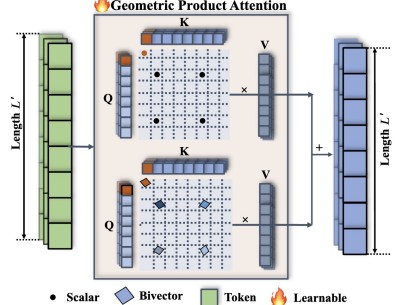

Figure 4: A simplified illustration of geometric product attention. The entries of the attention matrix are multi-vectors.

*C) Linear Projection.* Given time-frequency tokens $\boldsymbol{U}^{(s)} = \{\boldsymbol{u}_1^{(s)}, \boldsymbol{u}_2^{(s)}, \cdots, \boldsymbol{u}_{L'}^{(s)}\} \in \mathbb{R}^{C \times L'}$ for scale $s$ and shared weights $\boldsymbol{W}_Q, \boldsymbol{W}_K, \boldsymbol{W}_V \in \mathbb{R}^{L' \times L'}$, the query, key, and value matrices are:

$$\boldsymbol{Q}^{(s)} = \boldsymbol{U}^{(s)} \boldsymbol{W}_Q, \ \boldsymbol{K}^{(s)} = \boldsymbol{U}^{(s)} \boldsymbol{W}_K, \ \boldsymbol{V}^{(s)} = \boldsymbol{U}^{(s)} \boldsymbol{W}_V, \text{ for scale } s \in \{0, 1, \cdots, S\}. \quad (4)$$

To keep the number of channels/variables unchanged, we apply the linear projection along $L'$.

*D) Geometric attention calculation.* Consider the expression $\boldsymbol{\alpha} \cdot \boldsymbol{\beta} + \boldsymbol{\alpha} \wedge \boldsymbol{\beta}$ and let us evaluate how we can minimally modify the self-attention block to mimic this behavior. We can consider two different $\boldsymbol{V}^{(s)}$'s: say $\boldsymbol{V}_1^{(s)}$ and $\boldsymbol{V}_2^{(s)}$. The dot-product attention between the tokens can act upon $\boldsymbol{V}_1^{(s)}$ for the term $\boldsymbol{Q}^{(s)^T} \boldsymbol{K}^{(s)}$. Separately, the matrix of wedge-product objects acts upon $\boldsymbol{V}_2^{(s)}$ for the second term $\boldsymbol{B} = \{\boldsymbol{B}_{tt'}\}$ for $t, t' \in \{1, \cdots, L'\}$ with $\boldsymbol{B}_{tt'} = \boldsymbol{q}_t^{(s)} \wedge \boldsymbol{k}_{t'}^{(s)}$.

The first part can be viewed simply as the vanilla attention mechanism, so no special treatment is needed. The wedge product results in a matrix of bivector objects, where each element $\boldsymbol{B}_{tt'}$ is a bivector for the pair-wise tokens; we indeed compute it element-wise for each pair $t$ and $t'$. Viewing $\boldsymbol{V}_2^{(s)}$ column by column, the operation $\boldsymbol{B}\boldsymbol{V}_2^{(s)}$ is well defined and is closed within the algebra. For example, $\boldsymbol{B}$ is an $L' \times L'$ matrix where each element $\boldsymbol{B}_{tt'}$ is a bivector resulting from the wedge product of the $t$-th query vector and the $t'$-th key vector, while $\boldsymbol{V}_2^{(s)}$ is an $L' \times C$ matrix. The operation can be written explicitly as the sum over the geometric product between a bivector $\boldsymbol{B}_{tt'}$ and entries from a column of $\boldsymbol{V}_2^{(s)}$. The only remaining task is to combine this result with vanilla self-attention, and to do so, we need to map these bivectors down. For this, we use a reduction function $\zeta(\cdot)$ to match dimensions. There is much flexibility in choosing $\zeta(\cdot)$: it can be the bivector's magnitude or a trainable MLP that takes both magnitude and orientation as an input.

**Summary of geometric product attention mechanism.** The geometric attention mechanism is:

$$\text{GeoProdAttn}(\boldsymbol{Q}, \boldsymbol{K}, \boldsymbol{V}) = \text{softmax}\left(\frac{\text{dot-prod}(\boldsymbol{Q}, \boldsymbol{K})}{\sqrt{C}}\right)\boldsymbol{V} + \zeta\left(\left(\frac{\text{wedge-prod}(\boldsymbol{Q}, \boldsymbol{K})}{\sqrt{C}}\right)\boldsymbol{V}\right) \tag{5}$$

where $C$ is a scaling factor and we have used $\boldsymbol{V}$ instead of two separate variables. Also, the matrix of bivectors acts upon $\boldsymbol{V}$ individually for each column in $\boldsymbol{V}$.

## 5 MODULE 3: RECONSTRUCTION OF MULTIVARIATE TIME SERIES

**Motivation/Rationale.** After processing the time-frequency tokens through the geometric product attention module described above, we need to reconstruct the signal in the time domain. This is achieved using a learnable $\text{ISWT}(\cdot; \boldsymbol{h}_1, \boldsymbol{g}_1)$, where $\boldsymbol{h}_1$ and $\boldsymbol{g}_1$ are the learnable synthesis filters for the low-pass and high-pass components, respectively. These filters are the direct counterparts to the analysis filters $\boldsymbol{h}_0$ and $\boldsymbol{g}_0$ used in the forward SWT.

**Details of the construction.** The reconstruction is performed iteratively, starting from the coarsest scale and progressing to the finest scale. Given the initial approximation coefficients $\hat{\boldsymbol{a}}^{(S)} = \hat{\boldsymbol{u}}^{(0)}$ and the processed tokens at each scale $s$, denoted as $\{\hat{\boldsymbol{u}}_1^{(s)}, \hat{\boldsymbol{u}}_2^{(s)}, \ldots, \hat{\boldsymbol{u}}_{L'}^{(s)}\}_{s=0}^{S}$, the reconstruction can be written as:

$$\hat{\boldsymbol{a}}_t^{(s-1)} = \sum_k \boldsymbol{h}_1^{(s)}(k)\hat{\boldsymbol{a}}_{t+k}^{(s)} + \sum_k \boldsymbol{g}_1^{(s)}(k)\hat{\boldsymbol{u}}_{t+k}^{(s)}. \tag{6}$$

where $\boldsymbol{h}_1^{(s)}$ and $\boldsymbol{g}_1^{(s)}$ are the upsampled versions of $\boldsymbol{h}_1$ and $\boldsymbol{g}_1$ at level $s$. The reconstruction process iteratively computes $\hat{\boldsymbol{a}}^{(s-1)}$ using $\hat{\boldsymbol{a}}^{(s)}$ and $\hat{\boldsymbol{u}}^{(s)}$ until we reach $\hat{\boldsymbol{a}}^{(0)}$.

**Summary of reconstruction.** The final reconstructed time series $\hat{\boldsymbol{X}} = \{\hat{\boldsymbol{x}}_1, \hat{\boldsymbol{x}}_2, \cdots, \hat{\boldsymbol{x}}_{L'}\}$ is given by $\hat{\boldsymbol{a}}^{(0)}$. This reconstructed representation preserves the temporal structure of the original input while incorporating the multi-scale information processed via geometric product attention. This reconstructed time-domain representation $\hat{\boldsymbol{X}}$ is then passed through a feed-forward network and layer normalization for final refinement, which produces the forecast output to calculate the loss. We perform end-to-end training.

## 6 EXPERIMENT

In this section, we cover our experimental findings in detail. We divide our experimental protocol into two phases: evaluating the quality of forecasting both for long-term and short-term and an ablation study to evaluate the efficacy of our proposed model, SimpleTM.

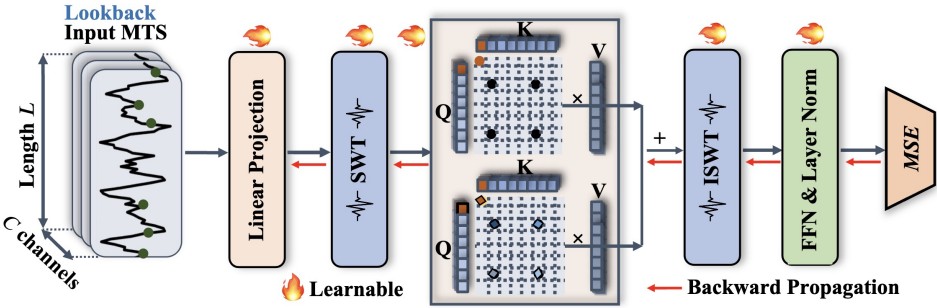

Figure 5: An overview of the proposed SimpleTM model framework.

## 6.1 SETUP AND BASELINES

**Baselines.** We compare our model with 15 well-known forecasting models for MTS data, including

**(a)** *MLP-based methods*: TimeMixer Wang et al. (2024), TiDE Das et al. (2023), RLinear Li et al. (2024), DLinear Zeng et al. (2023);

**(b)** *Transformer-based methods*: iTransformer Liu et al. (2024a), PatchTST Nie et al. (2023), Crossformer Zhang & Yan (2023), FEDformer Zhou et al. (2022b), Autoformer Wu et al. (2021), FiLMZhou et al. (2022a), StationaryLiu et al. (2022b);

**(c)** *CNN-based methods*: TimesNet Wu et al. (2023), SCINet LIU et al. (2022), MICN Wang et al. (2023);

**(d)** *GNN-based method*: CrossGNN Huang et al. (2023).

**Datasets.** The datasets that are covered in our experiments include:

**(a)** *Long-term forecasting*: We evaluate our model on 8 widely recognized benchmarks: the ETT datasets (**ETTh**1, **ETTh**2, **ETTm**1, and **ETTm**2), which provides seven factors of electricity transformer data recorded at hourly and 15-minute intervals, as well as the **Weather**, **Solar-Energy**, **Electricity**, and **Traffic** datasets, which include diverse meteorological, power production, consumption, and road occupancy data Wu et al. (2021).

**(b)** *Short-term forecasting*: We adopt the PEMS dataset Chen et al. (2001) with four public traffic subsets (**PEMS03**, **PEMS04**, **PEMS07**, and **PEMS08**) recorded every 5 minutes. We also evaluate the forecastability of all datasets, following Wang et al. (2024), and observe that ETT and Solar-Energy pose modeling challenges due to their low forecastability. Further details on the datasets can be found in Appendix A.

## 6.2 EVALUATION RESULTS

**Long-term forecasting results:** Forecast results from our experiments are presented in Table 1, with optimal performance denoted in **red** and second-best in blue. A lower MSE/MAE values indicates superior prediction accuracy. SimpleTM demonstrates robust performance across diverse benchmarks, achieving optimal MSE/MAE in 7 out of 8 datasets. Complete results are in Appendix B. We briefly discuss comparisons with two of the closest methods in terms of performance.

**(a)** *TimeMixer Wang et al. (2024)*: Our method exhibits MSE reductions of 8.1% for ETTh2 and 8.8% for ECL compared to TimeMixer. In the Solar-Energy dataset, acknowledged for its complexity, our model attains the best MSE, surpassing TimeMixer by 14.8%. Although TimeMixer uses a multi-scale approach, it underperforms in high-dimensional datasets because its mixing mechanism is limited to linear or lower-order interactions. Its reliance on average pooling and ensemble predictions leads to information loss during scale transitions. SimpleTM overcomes these challenges with a geometric attention mechanism in $G2$ space, combined with a classical tokenization method, enabling it to capture complex, higher-order relationships by considering both the magnitude and orientation of token pairs.

Table 1: Long-term forecasting results for various prediction horizons $H \in \{96, 192, 336, 720\}$ with a fixed lookback window of $L = 96$. Reported values are averaged across prediction lengths.

| Model | SimpleTM (Ours) | | TimeMixer (2024) | | iTransformer (2024a) | | CrossGNN (2023) | | RLinear (2024) | | PatchTST (2023) | | Crossformer (2023) | | TiDE (2023) | | TimesNet (2023) | | DLinear (2023) | | SCINet (2022) | |
|---|---|---|---|---|---|---|---|---|---|---|---|---|---|---|---|---|---|---|---|---|---|---|
| Metric | MSE | MAE | MSE | MAE | MSE | MAE | MSE | MAE | MSE | MAE | MSE | MAE | MSE | MAE | MSE | MAE | MSE | MAE | MSE | MAE | MSE | MAE |
| ETTm1 | **0.381** | **0.396** | 0.385 | 0.399 | 0.407 | 0.410 | 0.393 | 0.404 | 0.414 | 0.407 | 0.387 | 0.400 | 0.513 | 0.496 | 0.419 | 0.419 | 0.400 | 0.406 | 0.403 | 0.407 | 0.485 | 0.481 |
| ETTm2 | **0.275** | **0.322** | 0.278 | 0.325 | 0.288 | 0.332 | 0.282 | 0.330 | 0.286 | 0.327 | 0.281 | 0.326 | 0.757 | 0.610 | 0.358 | 0.404 | 0.291 | 0.333 | 0.350 | 0.401 | 0.571 | 0.537 |
| ETTh1 | **0.422** | **0.428** | 0.458 | 0.445 | 0.454 | 0.447 | 0.437 | 0.434 | 0.446 | 0.434 | 0.469 | 0.454 | 0.529 | 0.522 | 0.541 | 0.507 | 0.458 | 0.450 | 0.456 | 0.452 | 0.747 | 0.647 |
| ETTh2 | **0.353** | **0.391** | 0.384 | 0.407 | 0.383 | 0.407 | 0.393 | 0.413 | 0.374 | 0.398 | 0.387 | 0.407 | 0.942 | 0.684 | 0.611 | 0.550 | 0.414 | 0.427 | 0.559 | 0.515 | 0.954 | 0.723 |
| ECL | **0.166** | **0.260** | 0.182 | 0.272 | 0.178 | 0.270 | 0.201 | 0.300 | 0.219 | 0.298 | 0.205 | 0.290 | 0.244 | 0.334 | 0.251 | 0.344 | 0.192 | 0.295 | 0.212 | 0.300 | 0.268 | 0.365 |
| Traffic | 0.444 | 0.289 | 0.484 | 0.297 | **0.428** | **0.282** | 0.583 | 0.323 | 0.626 | 0.378 | 0.481 | 0.304 | 0.550 | 0.304 | 0.760 | 0.473 | 0.620 | 0.336 | 0.625 | 0.383 | 0.804 | 0.509 |
| Weather | **0.243** | **0.271** | 0.245 | 0.276 | 0.258 | 0.278 | 0.247 | 0.289 | 0.272 | 0.291 | 0.259 | 0.281 | 0.259 | 0.315 | 0.271 | 0.320 | 0.259 | 0.287 | 0.265 | 0.317 | 0.292 | 0.363 |
| Solar-Energy | **0.184** | **0.247** | 0.216 | 0.280 | 0.233 | 0.262 | 0.249 | 0.313 | 0.369 | 0.356 | 0.270 | 0.307 | 0.641 | 0.639 | 0.347 | 0.417 | 0.301 | 0.319 | 0.330 | 0.401 | 0.282 | 0.375 |

Table 2: Short-term forecasting results on PEMS datasets. Results are shown for prediction horizon $H = 12$ with a fixed lookback window of $L = 96$. Lower metric values indicate more accurate predictions.

| | Model Metric | SimpleTM (Ours) | iTransformer (2024a) | TimeMixer (2024) | Crossformer (2023) | PatchTST (2023) | TimesNet (2023) | MICN (2023) | DLinear (2023) | FiLM (2022a) | FEDformer (2022b) | Stationary (2022b) | Autoformer (2021) |
|---|---|---|---|---|---|---|---|---|---|---|---|---|---|
| PEMS03 | MAE | 14.86 | 18.13 | **14.80** | 15.64 | 18.95 | 16.41 | 15.71 | 19.70 | 21.36 | 19.00 | 17.64 | 18.08 |
| | MAPE | **14.79** | 19.19 | 14.79 | 15.74 | 17.29 | 15.17 | 15.67 | 18.35 | 18.35 | 18.57 | 17.56 | 18.75 |
| | RMSE | **23.58** | 28.86 | 23.58 | 25.56 | 30.15 | 26.72 | 24.55 | 32.35 | 35.07 | 30.05 | 28.37 | 27.82 |
| PEMS04 | MAE | **18.74** | 23.42 | 18.97 | 20.38 | 24.86 | 21.63 | 21.62 | 24.62 | 26.74 | 26.51 | 22.34 | 25.00 |
| | MAPE | **12.11** | 17.83 | 12.24 | 12.84 | 16.65 | 13.15 | 13.53 | 16.12 | 16.46 | 16.76 | 14.85 | 16.70 |
| | RMSE | **30.46** | 35.75 | 30.70 | 32.41 | 40.46 | 34.90 | 34.39 | 39.51 | 42.86 | 41.81 | 35.47 | 38.02 |
| PEMS07 | MAE | **20.25** | 22.54 | 20.76 | 22.54 | 27.87 | 25.12 | 22.28 | 28.65 | 28.76 | 27.92 | 26.02 | 26.92 |
| | MAPE | **8.55** | 12.77 | 8.77 | 9.38 | 12.69 | 10.60 | 9.57 | 12.15 | 11.21 | 12.29 | 11.75 | 11.83 |
| | RMSE | **33.06** | 33.92 | 33.71 | 35.49 | 42.56 | 40.71 | 35.40 | 45.02 | 45.85 | 42.29 | 42.34 | 40.60 |
| PEMS08 | MAE | **14.92** | 18.79 | 15.26 | 17.56 | 20.35 | 19.01 | 17.76 | 20.26 | 22.11 | 20.56 | 19.29 | 20.47 |
| | MAPE | **9.36** | 12.19 | 9.71 | 10.92 | 13.15 | 11.83 | 10.76 | 12.09 | 12.81 | 12.41 | 12.21 | 12.27 |
| | RMSE | **23.80** | 28.86 | 24.35 | 27.21 | 31.04 | 30.65 | 27.26 | 32.38 | 35.13 | 32.97 | 38.62 | 31.52 |

**(b)** *iTransformer Liu et al. (2024a)*: Against iTransformer, our method achieves MSE reductions of $6.4\%$, $7.0\%$, and $4.7\%$ for ETTm1, ETTh1, and ECL respectively. While iTransformer excels on high-dimensional time-series datasets, such as Traffic (862 variables/channels), it struggles with the rapidly fluctuating ETT datasets due to its variate-wise tokenization, which fails to capture fine-grained local patterns and lacks sufficient inter-channel context in lower-dimensional scenarios. In contrast, SimpleTM uses wavelet-based tokens that prioritize intra-variable local interactions and effectively capture oscillatory patterns across multiple resolutions.

**Short-term forecasting results:** Table 2 presents the short-term forecasting results for the high dimensional PEMS datasets. This is evaluated using three metrics: Mean Absolute Error (MAE), Mean Absolute Percentage Error (MAPE), and Root Mean Square Error (RMSE), where lower values indicate better predictions. Our SimpleTM demonstrates superior or comparable performance across all four PEMS datasets (PEMS03, PEMS04, PEMS07, and PEMS08), consistently achieving the best results. These results validate our model's good performance for high-dimensional, short-term forecasting tasks, complementing its strong performance in long-term forecasting scenarios. Additional results for larger prediction horizons are provided in Table 10 in Appendix B.

## 6.3 MISCELLANEOUS ADDITIONAL ANALYSIS: ABLATIONS, WAVELETS

**Ablation Study.** We conducted an ablation study to evaluate two key architectural elements: geometric attention mechanism and the stationary wavelet transform. Table 3 summarizes the results across four datasets, with metric values averaged over four prediction horizons. The findings consistently demonstrate that both geometric attention and SWT contribute to our model's performance. Detailed results can be found in Appendix C.

**Filters in the wavelet decomposition.** We now briefly check the properties of the learned filters. We randomly initialized the wavelet basis with $\ell_2$ normalization and compared the resulting filters to the wavelet bank for the ETTh2 dataset, identifying the most similar ground truth wavelet for each learned wavelet. As shown in Fig 6a (left), the learned filters from random initialization occasionally approximated wavelet-like structures, displaying higher ampli-

Table 3: Ablation study results comparing model variations across datasets.

| Model | ETTh1 | | ETTm1 | | Solar | | Weather | |
|---|---|---|---|---|---|---|---|---|
| | MSE | MAE | MSE | MAE | MSE | MAE | MSE | MAE |
| SimpleTM | **0.422** | **0.428** | **0.381** | **0.396** | **0.184** | **0.247** | **0.243** | **0.271** |
| w/o Attn | 0.437 | 0.440 | 0.385 | 0.398 | 0.194 | 0.253 | 0.245 | 0.273 |
| w/o SWT | 0.432 | 0.435 | 0.386 | 0.398 | 0.246 | 0.289 | 0.247 | 0.274 |

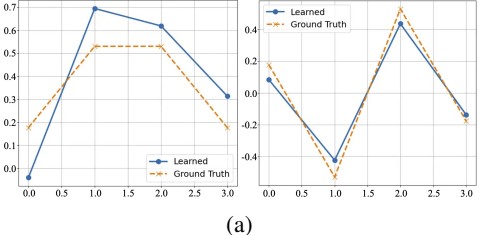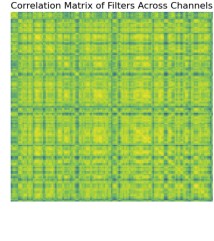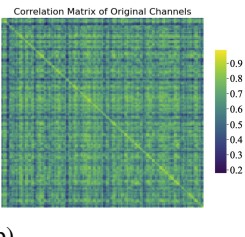

(a)                                                                                          (b)

Figure 6: Analysis of learned filters and their correlations in wavelet-based time series forecasting. (a) Comparison of learned filters with theoretical wavelet bases (Bior3.3). Left: synthesis low-pass filter; Right: forward high-pass filter. (b) Correlation heatmaps of learned filters (left) and original channels (right).

tude peaks and maintaining overall patterns, although with some amplitude variations. This suggests that the model can inherently discover wavelet-like features without explicit wavelet priors. In contrast, filters initialized with standard wavelets (the right in Fig 6a) retained their core structure while exhibiting subtle adaptations, indicating that a wavelet initialization provides a strong inductive bias for refining theoretically grounded filters based on empirical data. A comparison of the correlation heatmaps in Fig. 6b between the filters (all initialized with identical standard wavelets) and the original channels reveals meaningful patterns. The filter correlation matrix shows a distinct block-like structure with high correlations $(0.7 - 0.9)$, primarily due to shared initialization. However, dark horizontal and vertical lines suggest that some filters have developed lower correlations, indicating a degree of specialization. In contrast, the original channel correlation matrix shows weaker overall correlations $(0.4 - 0.6)$ with a less pronounced block structure. The persistence of some block-like patterns in both matrices, albeit at different scales, implies that the model retains aspects of the original data structure while enhancing certain relationships through learned filters.

**Multi-scale visualization.** Fig. 7 presents a representative example of our multiscale forecasting results. The MTS forecasting panel illustrates the model's effectiveness in accurately predicting overall patterns, including significant peaks and troughs, while capturing the cyclical nature of the data. Our model excels at decomposing and reconstructing the time series across multiple scales. The Scale 1 and Scale 2 panels highlight the model's ability to capture high-frequency fluctuations, while the Scale 0 panel reveals the underlying low-frequency trend. This multi-resolution analysis enables the model to extract relevant features from various timescales and integrate them into a coherent forecast in the original domain. In summary, our multiscale forecasting technique demonstrates robust performance in capturing both macro trends and micro fluctuations.

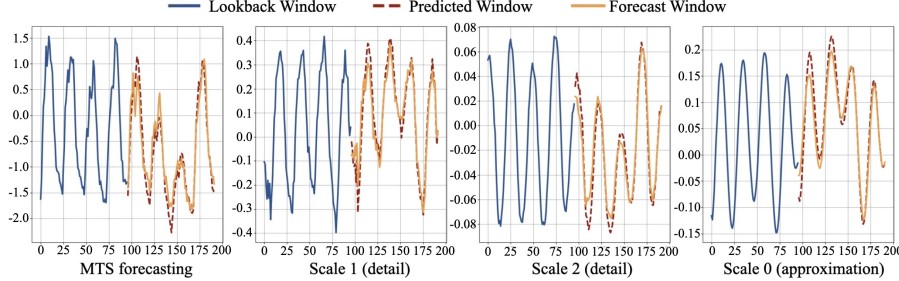

Figure 7: Multiscale forecasting visualization: MTS prediction shows global pattern and cyclical nature. Scale decomposition demonstrates the ability to capture low-frequency trends (Scale 0) and progressively higher-frequency fluctuations (Scales 1-2).

**Compute and Memory Footprint.** A key advantage of our single- or two-layer design is that it often captures complex dependencies in MTS data without resorting to larger, more cumbersome backbones. In our reported experiments, we prioritize memory and computation efficiency by choosing the bivector's magnitude for the reduction function $\zeta(\cdot)$. This approach not only matches or surpasses the accuracy of competitive baselines, but also reduces memory requirements and parameter counts. For instance, on the Weather dataset, our model uses only $0.3\%$ of iTransformer's parameters and $13\%$ of TimeMixer's parameters, requires $38\%$ and $66\%$ less memory, and runs $1.7x$ and $3.4\times$ faster, respectively. Further details are provided in Appendix D.

## 7 RELATED WORK

**Time Series Forecasting.** Time-series forecasting is a mature topic and has advanced from traditional statistical models like AIRMA (AutoRegressive Integrated Moving Average) Box & Jenkins (1994) and ARMA (AutoRegressive Moving Average) Markidakis & Hibon (1997) models to sophisticated deep learning approaches that better handle the complexity of time-series data. These approaches can be broadly categorized as follows.

1. *CNN models* effectively capture local temporal patterns in time series data. TCN Bai et al. (2018) introduced causal and dilated convolutions, while SCINet LIU et al. (2022) employed sample convolutions. TimesNet Wu et al. (2023) used 2D variation modeling with inception blocks to capture both inter-period and intra-period patterns. CNNs sometimes struggle with long-range forecasting due to their limited receptive field.

2. *Graph Neural Network (GNN)* methods are capable of capturing inter-variable relationships in MTS data. MTGNN Wu et al. (2020) used graph learning to infer variable interactions, and Cross-GNN Huang et al. (2023) further refined this with cross-scale and cross-variable modeling to manage noise in MTS data. However, GNNs can often be quite computationally intensive.

3. *MLP models* Zeng et al. (2023) offer a balance between simplicity and efficiency. TimeMixer Wang et al. (2024) introduced multi-scale mixing however average pooling in this context leads to some information loss when transferring between scales. RLinear Li et al. (2024) showed that linear models, with a careful design, could effectively capture periodic features, achieving competitive performance with more complex architectures.

4. *Transformer models* Wu et al. (2021); Zhou et al. (2021); Wu et al. (2022); Zhou et al. (2022b) are prominent results that have demonstrated efficacy in capturing long-range dependencies. Crossformer Zhang & Yan (2023) introduced cross-dimension self-attention, iTransformer Liu et al. (2024a) applied attention to channel-tokens but lacked the resolution to capture fine-grained local patterns and can struggle to gain sufficient inter-channel context, and PatchTST Nie et al. (2023) used a patch-based representation with channel-independent processing and a fixed resolution.

**Multi-scale Modeling.** Capturing patterns at different resolutions is very common in vision Fan et al. (2021); Lin et al. (2017); Tao et al. (2020) and has also been used to obtain efficient self-attention modules Nguyen et al. (2023); Zeng et al. (2022). Inspired by these successes, multi-scale modeling has been adapted to time-series forecasting as well. N-HiTS Challu et al. (2023) constructed a hierarchical forecast with multi-rate sampling, while Scaleformer Shabani et al. (2023) progressively refined forecasts through repeated upsampling and downsampling operations. Pathformer Chen et al. (2024) applied dual attention over patches of varying temporal size. Pyraformer Liu et al. (2022a) used a pyramidal attention structure to handle inter-scale dependencies. TimeMixer Wang et al. (2024) used a decomposable mixing approach, combining seasonal and trend components separately across scales for both past and future temporal variations. Different subsets of these methods face different challenges. Manually designed scales can make the model inflexible to adapt to dynamic time series, while average pooling often results in the loss of fine-grained details. In some models, the aggregation and reconstruction mechanisms are fragmented, requiring ensemble strategies or more complicated architectures.

## 8 CONCLUSIONS

Our work introduces a novel approach to multivariate time series (MTS) analysis that integrates a simple wavelet-based tokenization and a generalized form self-attention that captures both multi-scale temporal dynamics and complex inter-channel relationships. Our empirical results demonstrate competitive performance against most existing baselines across various MTS tasks. Our experiments suggest that exploiting inter-channel dependency does not always yield improvements, and the performance varies from one dataset to other other. The construction is simple and can provide a sensible lightweight baseline for more sophisticated methods although in its current form, cannot easily be extended to token-by-token generation. We must acknowledge that as MTS datasets that are publicly available grow in size, larger models may be better suited to maximize performance. Nonetheless, we believe that the individual components utilized in our formulation can still meaningfully inform the design of specialized adapters and/or embedding/tokenization schemes.

**Acknowledgments.** The authors are grateful to Sourav Pal, WonHwa Kim, Melanie Boly and Guilio Tononi for discussions and input. Partial support for this work came from a contract to UW-Madison under the DARPA Strengthen program.

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

## A  EXPERIMENT DETAILS

**Datasets.** We evaluate our model on several benchmark datasets covering both long-term and short-term forecasting tasks. Among long-term forecasting tasks, we evaluate our method on 1) ETT Zhou et al. (2021): a dataset of electricity transformer data, which includes four subsets (ETTh1, ETTh2, ETTm1, ETTm2). ETTh1 and ETTh2 record data hourly, while ETTm1 and ETTm2 record data every 15 minutes; 2) Weather Wu et al. (2021): a dataset comprising 21 meteorological parameters and is collected at a 10-minute interval; 3) Solar-Energy Lai et al. (2018): a dataset recording power generation data from multiple plants, with data collected every 10 minutes in 2006; 4) Electricity Wu et al. (2021): a dataset of electricity consumption for 321 clients; 5) Traffic Wu et al. (2021): a datasets monitoring hourly road occupancy rates through 862 sensors in San Francisco from 2015 to 2016. Additionally, we evaluate our method on PEMS dataset for short-term forecasting. The PEMS dataset collects traffic network data from various locations and covers four subsets (PEMS03, PEMS04, PEMS07, PEMS08), which has been widely adopted as benchmarks since LIU et al. (2022).

We mainly follow the experimental configurations in Wu et al. (2023), including the same data processing and splitting protocol. For both the long-term and short-term forecasting settings, we fix the lookback window length to 96 for all datasets and baselines. The prediction lengths vary according to the forecasting tasks: for ETT family, Weather, Solar-Energy, ECL, and Traffic datasets in the long-term forecasting task, we use prediction lengths of $\{96, 192, 336, 720\}$, while for the PEMS dataset in the short-term forecasting task, we employ prediction lengths of $\{12, 24, 48, 96\}$. Details of the dataset are provided in Table 4.

Table 4: Dataset statistics. The dimension indicates the number of channels/variates, and the dataset size is organized in (training, validation, testing).

| Tasks | Dataset | Dim. | Prediction Length | Dataset Size | Frequency | Domain |
|---|---|---|---|---|---|---|
| Long-term Forecasting | ETTm1 | 7 | $\{96, 192, 336, 720\}$ | (34465, 11521, 11521) | 15 min | Temperature |
| | ETTm2 | 7 | $\{96, 192, 336, 720\}$ | (34465, 11521, 11521) | 15 min | Temperature |
| | ETTh1 | 7 | $\{96, 192, 336, 720\}$ | (8545, 2881, 2881) | 1 hour | Temperature |
| | ETTh2 | 7 | $\{96, 192, 336, 720\}$ | (8545, 2881, 2881) | 1 hour | Temperature |
| | Electricity | 321 | $\{96, 192, 336, 720\}$ | (18317, 2633, 5261) | Hourly | Electricity |
| | Traffic | 862 | $\{96, 192, 336, 720\}$ | (12185, 1757, 3509) | Hourly | Transportation |
| | Weather | 21 | $\{96, 192, 336, 720\}$ | (36792, 5271, 10540) | 10 min | Weather |
| | Solar-Energy | 137 | $\{96, 192, 336, 720\}$ | (36601, 5161, 10417) | 10 min | Electricity |
| Shor-term Forecasting | PEMS03 | 358 | $\{12, 24, 48, 96\}$ | (15617, 5135, 5135) | 5 min | Transportation |
| | PEMS04 | 307 | $\{12, 24, 48, 96\}$ | (10172, 3375, 3375) | 5 min | Transportation |
| | PEMS07 | 883 | $\{12, 24, 48, 96\}$ | (16911, 5622, 5622) | 5 min | Transportation |
| | PEMS08 | 170 | $\{12, 24, 48, 96\}$ | (10690, 3548, 265) | 5 min | Transportation |

**Hyperparameter search.** Table 5 summarizes the hyperparameters and training settings used in our experiments. Our hyperparameter selection followed a systematic approach, combining grid search with domain-specific considerations. The number of layers was fixed at 1, and the input length $L$ was set to 96 to ensure fair comparisons across benchmark datasets. The pseudo length $L'$ was configured based on the input dimensionality of each dataset. Larger values ($L' = 256$) were assigned to datasets with more input channels, and smaller values ($L' = 32$) were used for datasets with fewer channels to balance computational efficiency with model capacity.

The selection of wavelet initialization types was guided by both systematic evaluation and signal processing principles. We explored common wavelet families $\{db1, db4, db8, db12, bior3.1\}$, with specific choices informed by the temporal characteristics (i.e., sampling frequency) of each dataset. Specifically, the db1 (Haar) wavelet was primarily employed for datasets exhibiting high total variation (e.g., hourly-sampled datasets such as ETTh1, ECL, Traffic) due to its effectiveness in capturing sharp transitions. Conversely, longer filters (bior3.1, db4, db8) were utilized for higher-frequency data (e.g., minute-level datasets such as ETTm2, Weather, Solar-Energy, PEMS04) to better capture their smoother temporal patterns. We acknowledge that any of them are suitable initializations. The scale parameter $S$ was fixed at 3.

For training parameters, we performed a grid search over learning rates within a logarithmic scale from $10^{-3}$ to $2 \times 10^{-2}$. Batch sizes and training epochs were systematically evaluated within the ranges $\{16, 24, 256\}$ and $\{10, 20\}$, respectively. Larger batch sizes (256) and fewer training epochs (10) were typically assigned to long-term forecasting tasks, while smaller batch sizes (16) and more training epochs (20) were used for short-term forecasting scenarios. This approach to hyperparameter optimization enabled us to achieve good performance while accounting for the distinct temporal and structural characteristics of each dataset.

**Fair comparison settings.** To ensure a fair comparison, we maintained a consistent lookback window length of 96 across all experiments. Our baseline comparisons mimic the experimental protocols established in TimesNet Wu et al. (2023), including identical data processing and splitting procedures. We applied early stopping to all baselines when the validation loss failed to decrease for three consecutive epochs. Recent baselines, such as iTransformer Liu et al. (2024a), TimeMixer Wang et al. (2024), and CrossGNN Huang et al. (2023), adopted the same fair comparison settings. Therefore, their experimental configurations required no modifications, and we utilized their official repositories directly for reproduction. For baselines published prior to 2024, we used the long-term forecasting results provided in the TimesNet Wu et al. (2023) repository. These results were built on the experimental configurations provided by each model's original paper or official code. We verified that all hyperparameters for these baselines were selected from their respective official repositories while ensuring consistency with the fair comparison settings, where the only change were the input and output sequence lengths of all baseline models. Additionally, we adopted all baselines' short-term forecasting results directly from TimeMixer Wang et al. (2024), consistent with the fair comparison settings established by TimesNet.

**Implementation Details.** All experiments were conducted using PyTorch Paszke et al. (2019) on a single NVIDIA A100 40GB GPU. The model was trained using the Adam optimizer Kingma & Ba (2015) with Mean Absolute Error (MAE) as the loss function specifically for the PEMS datasets, following TimeMixer Wang et al. (2024), and Mean Squared Error (MSE) as the loss function otherwise.

Table 5: Summary of the experimental configurations for all datasets with a prediction length of 96.

| Dataset / Configuration | Model Hyperparameter | | | | | Training Process | | | |
|---|---|---|---|---|---|---|---|---|---|
| | Layers | Input Length $L$ | Pseudo Length $L'$ | Wavelet Initialization | Scale $S$ | Learning Rate | Attention | Batch Size | Epochs |
| ETTh1 | 1 | 96 | 32 | db1 | 3 | $2 \times 10^{-2}$ | Geometric | 256 | 10 |
| ETTh2 | 1 | 96 | 32 | bior3.1 | 3 | $6 \times 10^{-3}$ | Geometric | 256 | 10 |
| ETTm1 | 1 | 96 | 32 | db1 | 3 | $2 \times 10^{-2}$ | Geometric | 256 | 10 |
| ETTm2 | 1 | 96 | 32 | bior3.1 | 3 | $6 \times 10^{-3}$ | Geometric | 256 | 10 |
| Weather | 1 | 96 | 32 | db4 | 3 | $10^{-2}$ | Geometric | 256 | 10 |
| Solar-Energy | 1 | 96 | 128 | db8 | 3 | $10^{-2}$ | Geometric | 256 | 10 |
| Electricity | 1 | 96 | 256 | db1 | 3 | $10^{-2}$ | Geometric | 256 | 10 |
| Traffic | 1 | 96 | 256 | db1 | 3 | $6 \times 10^{-3}$ | Geometric | 24 | 10 |
| PEMS03 | 1 | 96 | 256 | bior3.1 | 3 | $2 \times 10^{-3}$ | Geometric | 16 | 20 |
| PEMS04 | 1 | 96 | 256 | bior3.1 | 3 | $2 \times 10^{-3}$ | Geometric | 16 | 20 |
| PEMS07 | 1 | 96 | 256 | db12 | 3 | $2 \times 10^{-3}$ | Geometric | 16 | 20 |
| PEMS08 | 1 | 96 | 256 | db1 | 3 | $10^{-3}$ | Geometric | 16 | 20 |

# B   COMPLETE RESULTS OF FORECASTING TASKS

## B.1   FULL RESULTS

**Long-term forecasting task.** In the long-term forecasting results presented in Table 1 of the main paper, we reported only the averaged performance across four prediction lengths due to space constraints. Table 6 provides a comprehensive breakdown of empirical results for each prediction length. Within each row, the lowest MSE and MAE scores are highlighted in **red**, and the second-lowest scores are underscored in blue. Our proposed method consistently achieves near top-2 performance across all evaluations.

**Additional baselines in long-term forecasting.** We evaluated our approach against conventional statistical time series forecasting methods, specifically ARIMA and ETS models, using the ETTh1

Table 6: Complete results of the long-term forecasting task, with an input length of 96 for all tasks. The reported metrics include the averaged Mean Squared Error (MSE) and Mean Absolute Error (MAE) across four prediction horizons, where lower values indicate better model performance.

| Model | | | SimpleTM (Ours) | | TimeMixer (2024) | | iTransformer (2024) | | CrossGNN (2024) | | RLinear (2023) | | PatchTST (2023) | | Crossformer (2023) | | TiDE (2023) | | TimesNet (2023) | | DLinear (2023) | | SCINet (2022) | | FEDformer (2022) | | Stationary (2022) | | Autoformer (2021) | |
|---|---|---|---|---|---|---|---|---|---|---|---|---|---|---|---|---|---|---|---|---|---|---|---|---|---|---|---|---|---|---|
| Metric | | | MSE | MAE | MSE | MAE | MSE | MAE | MSE | MAE | MSE | MAE | MSE | MAE | MSE | MAE | MSE | MAE | MSE | MAE | MSE | MAE | MSE | MAE | MSE | MAE | MSE | MAE | MSE | MAE |
| ETTm1 | 96 | | 0.321 | 0.361 | 0.328 | 0.363 | 0.334 | 0.368 | 0.335 | 0.373 | 0.355 | 0.376 | 0.329 | 0.367 | 0.404 | 0.426 | 0.364 | 0.387 | 0.338 | 0.375 | 0.345 | 0.372 | 0.418 | 0.438 | 0.379 | 0.419 | 0.386 | 0.398 | 0.505 | 0.475 |
| | 192 | | 0.360 | 0.380 | 0.364 | 0.384 | 0.377 | 0.391 | 0.372 | 0.390 | 0.391 | 0.392 | 0.367 | 0.385 | 0.450 | 0.451 | 0.398 | 0.404 | 0.374 | 0.387 | 0.380 | 0.389 | 0.439 | 0.450 | 0.426 | 0.441 | 0.459 | 0.444 | 0.553 | 0.496 |
| | 336 | | 0.390 | 0.404 | 0.390 | 0.404 | 0.426 | 0.420 | 0.403 | 0.411 | 0.424 | 0.415 | 0.399 | 0.410 | 0.532 | 0.515 | 0.428 | 0.425 | 0.410 | 0.411 | 0.413 | 0.413 | 0.490 | 0.485 | 0.445 | 0.459 | 0.495 | 0.464 | 0.621 | 0.537 |
| | 720 | | 0.454 | 0.438 | 0.458 | 0.445 | 0.491 | 0.459 | 0.461 | 0.442 | 0.487 | 0.450 | 0.454 | 0.439 | 0.666 | 0.589 | 0.487 | 0.461 | 0.478 | 0.450 | 0.474 | 0.453 | 0.595 | 0.550 | 0.543 | 0.490 | 0.585 | 0.516 | 0.671 | 0.561 |
| | Avg | | 0.381 | 0.396 | 0.385 | 0.399 | 0.407 | 0.410 | 0.393 | 0.404 | 0.414 | 0.407 | 0.387 | 0.400 | 0.513 | 0.496 | 0.419 | 0.419 | 0.400 | 0.406 | 0.403 | 0.407 | 0.485 | 0.481 | 0.448 | 0.452 | 0.481 | 0.456 | 0.588 | 0.517 |
| ETTm2 | 96 | | 0.173 | 0.257 | 0.176 | 0.259 | 0.180 | 0.264 | 0.176 | 0.266 | 0.182 | 0.265 | 0.175 | 0.259 | 0.287 | 0.366 | 0.207 | 0.305 | 0.187 | 0.267 | 0.193 | 0.292 | 0.286 | 0.377 | 0.203 | 0.287 | 0.192 | 0.274 | 0.255 | 0.339 |
| | 192 | | 0.238 | 0.299 | 0.242 | 0.303 | 0.250 | 0.309 | 0.240 | 0.307 | 0.246 | 0.304 | 0.241 | 0.302 | 0.414 | 0.492 | 0.290 | 0.364 | 0.249 | 0.309 | 0.284 | 0.362 | 0.399 | 0.445 | 0.269 | 0.328 | 0.280 | 0.339 | 0.281 | 0.340 |
| | 336 | | 0.296 | 0.338 | 0.304 | 0.342 | 0.311 | 0.348 | 0.304 | 0.345 | 0.307 | 0.342 | 0.305 | 0.343 | 0.597 | 0.542 | 0.377 | 0.422 | 0.321 | 0.351 | 0.369 | 0.427 | 0.637 | 0.591 | 0.325 | 0.366 | 0.334 | 0.361 | 0.339 | 0.372 |
| | 720 | | 0.393 | 0.395 | 0.393 | 0.397 | 0.412 | 0.407 | 0.406 | 0.400 | 0.407 | 0.398 | 0.402 | 0.400 | 1.730 | 1.042 | 0.558 | 0.524 | 0.408 | 0.403 | 0.554 | 0.522 | 0.960 | 0.735 | 0.421 | 0.415 | 0.417 | 0.413 | 0.433 | 0.432 |
| | Avg | | 0.275 | 0.322 | 0.278 | 0.325 | 0.288 | 0.332 | 0.282 | 0.330 | 0.286 | 0.327 | 0.281 | 0.326 | 0.757 | 0.610 | 0.358 | 0.404 | 0.291 | 0.333 | 0.350 | 0.401 | 0.571 | 0.537 | 0.305 | 0.349 | 0.306 | 0.347 | 0.327 | 0.371 |
| ETTh1 | 96 | | 0.366 | 0.392 | 0.381 | 0.401 | 0.386 | 0.405 | 0.382 | 0.398 | 0.386 | 0.395 | 0.414 | 0.419 | 0.423 | 0.448 | 0.479 | 0.464 | 0.384 | 0.402 | 0.386 | 0.400 | 0.654 | 0.599 | 0.376 | 0.419 | 0.513 | 0.491 | 0.449 | 0.459 |
| | 192 | | 0.422 | 0.421 | 0.440 | 0.433 | 0.441 | 0.436 | 0.427 | 0.425 | 0.437 | 0.424 | 0.460 | 0.445 | 0.471 | 0.474 | 0.525 | 0.492 | 0.436 | 0.429 | 0.437 | 0.432 | 0.719 | 0.631 | 0.420 | 0.448 | 0.534 | 0.504 | 0.500 | 0.482 |
| | 336 | | 0.440 | 0.438 | 0.501 | 0.462 | 0.487 | 0.458 | 0.465 | 0.445 | 0.479 | 0.446 | 0.501 | 0.466 | 0.570 | 0.546 | 0.565 | 0.515 | 0.491 | 0.469 | 0.481 | 0.459 | 0.778 | 0.659 | 0.459 | 0.465 | 0.588 | 0.535 | 0.521 | 0.496 |
| | 720 | | 0.463 | 0.462 | 0.501 | 0.482 | 0.503 | 0.491 | 0.472 | 0.468 | 0.481 | 0.470 | 0.500 | 0.488 | 0.653 | 0.621 | 0.594 | 0.558 | 0.521 | 0.500 | 0.519 | 0.516 | 0.836 | 0.699 | 0.506 | 0.507 | 0.643 | 0.616 | 0.514 | 0.512 |
| | Avg | | 0.422 | 0.428 | 0.458 | 0.445 | 0.454 | 0.447 | 0.437 | 0.434 | 0.446 | 0.434 | 0.469 | 0.454 | 0.529 | 0.522 | 0.541 | 0.507 | 0.458 | 0.450 | 0.456 | 0.452 | 0.747 | 0.647 | 0.440 | 0.460 | 0.570 | 0.537 | 0.496 | 0.487 |
| ETTh2 | 96 | | 0.281 | 0.338 | 0.292 | 0.343 | 0.297 | 0.349 | 0.309 | 0.359 | 0.288 | 0.338 | 0.302 | 0.348 | 0.745 | 0.584 | 0.400 | 0.440 | 0.340 | 0.374 | 0.333 | 0.387 | 0.707 | 0.621 | 0.358 | 0.397 | 0.476 | 0.458 | 0.346 | 0.388 |
| | 192 | | 0.355 | 0.387 | 0.374 | 0.395 | 0.380 | 0.400 | 0.390 | 0.406 | 0.374 | 0.390 | 0.388 | 0.400 | 0.877 | 0.656 | 0.528 | 0.509 | 0.402 | 0.414 | 0.477 | 0.476 | 0.860 | 0.689 | 0.429 | 0.439 | 0.512 | 0.493 | 0.456 | 0.452 |
| | 336 | | 0.365 | 0.401 | 0.428 | 0.433 | 0.428 | 0.432 | 0.426 | 0.444 | 0.415 | 0.426 | 0.426 | 0.433 | 1.043 | 0.731 | 0.643 | 0.571 | 0.452 | 0.452 | 0.594 | 0.541 | 1.000 | 0.744 | 0.496 | 0.487 | 0.552 | 0.551 | 0.482 | 0.486 |
| | 720 | | 0.413 | 0.436 | 0.454 | 0.458 | 0.427 | 0.445 | 0.445 | 0.444 | 0.420 | 0.440 | 0.431 | 0.446 | 1.104 | 0.763 | 0.874 | 0.679 | 0.462 | 0.468 | 0.831 | 0.657 | 1.249 | 0.838 | 0.463 | 0.474 | 0.562 | 0.560 | 0.515 | 0.511 |
| | Avg | | 0.353 | 0.391 | 0.384 | 0.407 | 0.383 | 0.407 | 0.393 | 0.413 | 0.374 | 0.398 | 0.387 | 0.407 | 0.942 | 0.684 | 0.611 | 0.550 | 0.414 | 0.427 | 0.559 | 0.515 | 0.954 | 0.723 | 0.437 | 0.449 | 0.526 | 0.516 | 0.450 | 0.459 |
| ECL | 96 | | 0.141 | 0.235 | 0.153 | 0.244 | 0.148 | 0.240 | 0.173 | 0.275 | 0.201 | 0.281 | 0.181 | 0.270 | 0.219 | 0.314 | 0.237 | 0.329 | 0.168 | 0.272 | 0.197 | 0.282 | 0.247 | 0.345 | 0.193 | 0.308 | 0.169 | 0.273 | 0.201 | 0.317 |
| | 192 | | 0.151 | 0.247 | 0.166 | 0.256 | 0.162 | 0.253 | 0.195 | 0.288 | 0.201 | 0.283 | 0.188 | 0.274 | 0.231 | 0.322 | 0.236 | 0.330 | 0.184 | 0.289 | 0.196 | 0.285 | 0.257 | 0.355 | 0.201 | 0.315 | 0.182 | 0.286 | 0.222 | 0.334 |
| | 336 | | 0.173 | 0.267 | 0.184 | 0.275 | 0.178 | 0.269 | 0.206 | 0.300 | 0.215 | 0.298 | 0.204 | 0.293 | 0.246 | 0.337 | 0.249 | 0.344 | 0.198 | 0.300 | 0.209 | 0.301 | 0.269 | 0.369 | 0.214 | 0.329 | 0.200 | 0.304 | 0.231 | 0.338 |
| | 720 | | 0.201 | 0.293 | 0.226 | 0.313 | 0.225 | 0.317 | 0.231 | 0.335 | 0.257 | 0.331 | 0.246 | 0.324 | 0.280 | 0.363 | 0.284 | 0.373 | 0.220 | 0.320 | 0.245 | 0.333 | 0.299 | 0.390 | 0.246 | 0.355 | 0.222 | 0.321 | 0.254 | 0.361 |
| | Avg | | 0.166 | 0.260 | 0.182 | 0.272 | 0.178 | 0.270 | 0.201 | 0.300 | 0.219 | 0.298 | 0.205 | 0.290 | 0.244 | 0.334 | 0.251 | 0.344 | 0.192 | 0.295 | 0.212 | 0.300 | 0.268 | 0.365 | 0.214 | 0.327 | 0.193 | 0.296 | 0.227 | 0.338 |
| Traffic | 96 | | 0.410 | 0.274 | 0.464 | 0.289 | 0.395 | 0.268 | 0.570 | 0.310 | 0.649 | 0.389 | 0.462 | 0.295 | 0.522 | 0.290 | 0.805 | 0.493 | 0.593 | 0.321 | 0.650 | 0.396 | 0.788 | 0.499 | 0.587 | 0.366 | 0.612 | 0.338 | 0.613 | 0.388 |
| | 192 | | 0.430 | 0.280 | 0.477 | 0.292 | 0.417 | 0.276 | 0.577 | 0.321 | 0.601 | 0.366 | 0.466 | 0.296 | 0.530 | 0.293 | 0.756 | 0.474 | 0.617 | 0.336 | 0.598 | 0.370 | 0.789 | 0.505 | 0.604 | 0.373 | 0.613 | 0.340 | 0.616 | 0.382 |
| | 336 | | 0.449 | 0.290 | 0.500 | 0.305 | 0.433 | 0.283 | 0.588 | 0.324 | 0.609 | 0.369 | 0.482 | 0.304 | 0.558 | 0.305 | 0.762 | 0.477 | 0.629 | 0.336 | 0.605 | 0.373 | 0.797 | 0.508 | 0.621 | 0.383 | 0.618 | 0.328 | 0.622 | 0.337 |
| | 720 | | 0.486 | 0.309 | 0.548 | 0.313 | 0.467 | 0.302 | 0.597 | 0.337 | 0.647 | 0.387 | 0.514 | 0.322 | 0.589 | 0.328 | 0.719 | 0.449 | 0.640 | 0.350 | 0.645 | 0.394 | 0.841 | 0.523 | 0.626 | 0.382 | 0.653 | 0.355 | 0.660 | 0.408 |
| | Avg | | 0.444 | 0.289 | 0.497 | 0.300 | 0.428 | 0.282 | 0.583 | 0.323 | 0.626 | 0.378 | 0.481 | 0.304 | 0.550 | 0.304 | 0.760 | 0.473 | 0.620 | 0.336 | 0.625 | 0.383 | 0.804 | 0.509 | 0.610 | 0.376 | 0.624 | 0.340 | 0.628 | 0.379 |
| Weather | 96 | | 0.162 | 0.207 | 0.165 | 0.212 | 0.174 | 0.214 | 0.159 | 0.218 | 0.192 | 0.232 | 0.177 | 0.218 | 0.158 | 0.230 | 0.202 | 0.261 | 0.172 | 0.220 | 0.196 | 0.255 | 0.221 | 0.306 | 0.217 | 0.296 | 0.173 | 0.223 | 0.266 | 0.336 |
| | 192 | | 0.208 | 0.248 | 0.209 | 0.253 | 0.221 | 0.254 | 0.211 | 0.266 | 0.240 | 0.271 | 0.225 | 0.259 | 0.206 | 0.277 | 0.242 | 0.298 | 0.219 | 0.261 | 0.237 | 0.296 | 0.261 | 0.340 | 0.276 | 0.336 | 0.245 | 0.285 | 0.307 | 0.367 |
| | 336 | | 0.263 | 0.290 | 0.264 | 0.293 | 0.278 | 0.296 | 0.267 | 0.310 | 0.292 | 0.307 | 0.278 | 0.297 | 0.272 | 0.335 | 0.287 | 0.335 | 0.280 | 0.306 | 0.283 | 0.335 | 0.309 | 0.378 | 0.339 | 0.380 | 0.321 | 0.338 | 0.359 | 0.395 |
| | 720 | | 0.340 | 0.341 | 0.342 | 0.345 | 0.358 | 0.347 | 0.352 | 0.362 | 0.364 | 0.353 | 0.354 | 0.348 | 0.398 | 0.418 | 0.351 | 0.386 | 0.365 | 0.359 | 0.345 | 0.381 | 0.377 | 0.427 | 0.403 | 0.428 | 0.414 | 0.410 | 0.419 | 0.428 |
| | Avg | | 0.243 | 0.271 | 0.245 | 0.276 | 0.258 | 0.278 | 0.247 | 0.289 | 0.272 | 0.291 | 0.259 | 0.281 | 0.259 | 0.315 | 0.271 | 0.320 | 0.259 | 0.287 | 0.265 | 0.317 | 0.292 | 0.363 | 0.309 | 0.360 | 0.288 | 0.314 | 0.338 | 0.382 |
| SolarEnergy | 96 | | 0.163 | 0.232 | 0.215 | 0.294 | 0.203 | 0.237 | 0.222 | 0.301 | 0.322 | 0.339 | 0.234 | 0.286 | 0.310 | 0.331 | 0.312 | 0.399 | 0.250 | 0.292 | 0.290 | 0.378 | 0.237 | 0.344 | 0.242 | 0.342 | 0.215 | 0.249 | 0.884 | 0.711 |
| | 192 | | 0.182 | 0.247 | 0.237 | 0.275 | 0.233 | 0.261 | 0.246 | 0.307 | 0.359 | 0.356 | 0.267 | 0.310 | 0.734 | 0.725 | 0.339 | 0.416 | 0.296 | 0.318 | 0.320 | 0.398 | 0.280 | 0.380 | 0.285 | 0.380 | 0.254 | 0.272 | 0.834 | 0.692 |
| | 336 | | 0.193 | 0.257 | 0.252 | 0.298 | 0.248 | 0.273 | 0.263 | 0.324 | 0.397 | 0.369 | 0.290 | 0.315 | 0.750 | 0.735 | 0.368 | 0.430 | 0.319 | 0.330 | 0.353 | 0.415 | 0.304 | 0.389 | 0.282 | 0.376 | 0.290 | 0.296 | 0.941 | 0.723 |
| | 720 | | 0.199 | 0.252 | 0.244 | 0.293 | 0.249 | 0.275 | 0.265 | 0.318 | 0.397 | 0.356 | 0.289 | 0.317 | 0.769 | 0.765 | 0.370 | 0.425 | 0.338 | 0.337 | 0.356 | 0.413 | 0.308 | 0.388 | 0.357 | 0.427 | 0.285 | 0.295 | 0.882 | 0.717 |
| | Avg | | 0.184 | 0.247 | 0.237 | 0.290 | 0.233 | 0.262 | 0.249 | 0.313 | 0.369 | 0.356 | 0.270 | 0.307 | 0.641 | 0.639 | 0.347 | 0.417 | 0.301 | 0.319 | 0.330 | 0.401 | 0.282 | 0.375 | 0.291 | 0.381 | 0.261 | 0.381 | 0.885 | 0.711 |

dataset. The evaluation covered four prediction horizons $\{96, 192, 336, 720\}$, with standardized preprocessing procedures applied across all models to ensure a fair comparison. Our proposed model consistently outperformed these methods across all prediction horizons, as shown in Table 7. This performance gap widened with increasing prediction lengths, aligning with our observations of these models. Statistical methods like ARIMA and ETS require a sufficient lookback period for robust parameter estimation and struggle with longer forecasting lengths due to error accumulation.

Table 7: Performance comparison of additional statistical models on the ETTh1 dataset across varying prediction lengths.

| Model | ETTh1-96 | | ETTh1-192 | | ETTh1-336 | | ETTh1-720 | |
|---|---|---|---|---|---|---|---|---|
| | MSE | MAE | MSE | MAE | MSE | MAE | MSE | MAE |
| SimpleTM | 0.366 | 0.392 | 0.422 | 0.421 | 0.440 | 0.438 | 0.463 | 0.462 |
| ETS | 1.145 | 0.658 | 1.185 | 0.855 | 1.234 | 0.963 | 2.298 | 1.818 |
| ARIMA | 1.010 | 0.719 | 1.033 | 0.635 | 1.204 | 0.700 | 2.269 | 1.072 |

We also compare our model's performance with zero-shot LLM Results Liu et al. (2024b); Woo et al. (2024); Goswami et al. (2024). We refer to the zero-shot performance reported by TIMER Liu et al. (2024b). However, because LLM-based models often uses a substantially larger context window (say 672 steps) than our 96-step lookback, direct comparison is inherently limited. To obtain a more rigorous assessment, one would need to match context windows and also evaluate fine-tuned versions of the LLM, an effort that would require significant computational resources. Consequently, we focus on demonstrating that a specialized model trained for a fixed-horizon forecasting task can still achieve strong performance, as shown in Table 8.

**Short-term forecasting task.** For short-term forecasting, we conducted additional comparisons with TimeMixer Wang et al. (2024), the previous state-of-the-art model, using its official repository and experimental configuration. The evaluation covered four prediction lengths $\{12, 24, 48, 96\}$. Table 10 presents the averaged RMSE/MAE values, along with their pooled standard deviations. Our model demonstrates consistent and statistically significant improvements across all PEMS datasets,

Table 8: Performance comparison of additional LLM-based models with prediction horizon of 96.

| Dataset | SimpleTM | TIMER-1B | TIMER-16B | TIMER-28B | MOIRAI-S | MOIRAI-M | MOIRAI-L | MOMENT |
|---------|----------|----------|-----------|-----------|----------|----------|----------|--------|
| ETTh1   | 0.366    | 0.438    | 0.364     | 0.393     | 0.441    | 0.383    | 0.394    | 0.674  |
| ETTh2   | 0.281    | 0.314    | 0.294     | 0.308     | 0.295    | 0.295    | 0.293    | 0.330  |
| ETTm1   | 0.321    | 0.690    | 0.766     | 0.420     | 0.562    | 0.448    | 0.452    | 0.670  |
| ETTm2   | 0.173    | 0.213    | 0.234     | 0.247     | 0.218    | 0.225    | 0.214    | 0.257  |
| ECL     | 0.141    | 0.192    | 0.139     | 0.147     | 0.212    | 0.162    | 0.155    | 0.744  |
| Traffic | 0.428    | 0.458    | 0.399     | 0.414     | 0.616    | 0.425    | 0.399    | 1.293  |
| Weather | 0.162    | 0.181    | 0.203     | 0.243     | 0.195    | 0.197    | 0.221    | 0.255  |

with error reductions ranging from $5.4\%$ to $17.4\%$. The most substantial improvements were observed on PEMS08, where our model reduced RMSE by $15.8\%$ and MAE by $17.4\%$. Notably, our model also shows more stable performance, as evidenced by the considerably smaller pooled standard deviations across all metrics and datasets.

### B.2 STABILITY ANALYSIS

**Pooled standard deviation.** The pooled standard deviation is calculated as

$$\bar{sd} = \sqrt{\frac{\sum_{i=1}^{4} \sum_{j=1}^{n} (x_{ij} - \bar{x}_i)^2}{4 \times (n-1)}},$$

where $n$ is the repeat times, $n-1$ is the degree of freedom within each prediction length, $i$ indexes the prediction lengths, $j$ indexes the repeats, $x_{ij}$ represents individual measurements, and $\bar{x}_i$ is the mean of repeats for each prediction length.

**Significance test.** To establish statistical significance, we used a Type II ANOVA analysis to assess the model effects (our proposed model versus other baseline model) while accounting for prediction length variations. The blocking design for prediction length effectively removed this source of variation from our error term, and increased statistical power to detect genuine differences between model architectures. The $p$-values reported in Table 9 test the null hypothesis that *there is no difference in performance between the proposed model and the baseline model.*

**Results.** We compare our proposed model with the second-best linear-based model, TimeMixer Wang et al. (2024) and third-best transformer-based model, iTransformer Liu et al. (2024a) across three repeats and four prediction lengths for both long-term and short-term forecasting tasks. As shown in Table 10, the pooled standard deviations are consistently smaller across all datasets, indicating the stability of our model's performance regardless of initialization. The consistently low p-values ($p < 0.05$) across all datasets confirm that the superior performance of our model is statistically significant and not attributable to random chance or prediction length variability. This is further supported by our additional short-term forecasting results with extended prediction lengths $\{12, 24, 48, 96\}$.

## C ABLATION STUDY

### C.1 ABLATIONS ON ARCHITECTURAL COMPONENTS

To rigorously validate our approach, we conducted additional experiments across four datasets (ETTh1, ETTm1, Weather, Solar-Energy) with four prediction lengths, each repeated three times. Through systematic component ablation, we evaluated two key architectural elements: geometric attention mechanism and the stationary wavelet transform.

**Geometric attention mechanism.** For geometric attention, we performed a direct comparison with vanilla attention. While not every dataset benefits equally–depending on the degree of cross-talk between channels–our findings show consistent performance improvements. For example, we observed a $3.55\%$ MSE reduction on ETTh1 and a $5.43\%$ reduction on Solar-Energy, all achieved

Table 9: Performance comparison of models on long-term forecasting tasks. The table reports the averaged Mean Squared Error (MSE) and Mean Absolute Error (MAE) across four prediction lengths, along with their pooled standard deviations (SD). Lower values indicate better model performance.

| Dataset | Model | MSE (Pooled SD) | MAE (Pooled SD) | MSE $p$-value | MAE $p$-value |
|---|---|---|---|---|---|
| ECL | SimpleTM | 0.166 (0.0008) | 0.260 (0.0006) | - | - |
| | TimeMixer | 0.182 (0.0012) | 0.272 (0.0006) | 0.000 | 0.000 |
| | iTransformer | 0.175 (0.0009) | 0.267 (0.0008) | 0.000 | 0.000 |
| ETTh1 | SimpleTM | 0.422 (0.0015) | 0.428 (0.0007) | - | - |
| | TimeMixer | 0.456 (0.0111) | 0.444 (0.0071) | 0.000 | 0.000 |
| | iTransformer | 0.456 (0.0035) | 0.448 (0.0024) | 0.000 | 0.000 |
| ETTh2 | SimpleTM | 0.353 (0.0021) | 0.391 (0.0015) | - | - |
| | TimeMixer | 0.386 (0.0074) | 0.407 (0.0043) | 0.000 | 0.000 |
| | iTransformer | 0.384 (0.0017) | 0.407 (0.0010) | 0.000 | 0.000 |
| ETTm1 | SimpleTM | 0.381 (0.0009) | 0.396 (0.0008) | - | - |
| | TimeMixer | 0.385 (0.0048) | 0.399 (0.0032) | 0.022 | 0.003 |
| | iTransformer | 0.408 (0.0012) | 0.412 (0.0010) | 0.000 | 0.000 |
| ETTm2 | SimpleTM | 0.275 (0.0012) | 0.322 (0.0011) | - | - |
| | TimeMixer | 0.278 (0.0026) | 0.325 (0.0018) | 0.001 | 0.000 |
| | iTransformer | 0.292 (0.0011) | 0.335 (0.0010) | 0.000 | 0.000 |
| Solar | SimpleTM | 0.184 (0.0016) | 0.247 (0.0031) | - | - |
| | TimeMixer | 0.237 (0.0088) | 0.290 (0.0242) | 0.000 | 0.000 |
| | iTransformer | 0.235 (0.0032) | 0.262 (0.0010) | 0.000 | 0.000 |
| Traffic | SimpleTM | 0.440 (0.0013) | 0.292 (0.0003) | - | - |
| | TimeMixer | 0.497 (0.0087) | 0.300 (0.0029) | 0.000 | 0.000 |
| | iTransformer | 0.422 (0.0008) | 0.282 (0.0005) | 0.000 | 0.000 |
| Weather | SimpleTM | 0.243 (0.0005) | 0.271 (0.0007) | - | - |
| | TimeMixer | 0.245 (0.0012) | 0.275 (0.0019) | 0.000 | 0.000 |
| | iTransformer | 0.261 (0.0023) | 0.281 (0.0021) | 0.000 | 0.000 |

Table 10: Performance comparison of models on short-term forecasting tasks. The table reports the averaged Root Mean Squared Error (RMSE) and Mean Absolute Error (MAE) across four prediction lengths $H \in \{12, 24, 48, 96\}$, along with their pooled standard deviations (SD). Lower values indicate better model performance.

| Dataset | Model | RMSE (Pooled SD) | MAE (Pooled SD) | RMSE $p$-value | MAE $p$-value |
|---|---|---|---|---|---|
| PEMS03 | SimpleTM | 29.08 (0.154) | 17.96 (0.065) | - | - |
| | TimeMixer | 31.73 (0.529) | 19.22 (0.278) | 0.000 | 0.000 |
| PEMS04 | SimpleTM | 32.91 (0.121) | 20.34 (0.077) | - | - |
| | TimeMixer | 34.78 (0.472) | 21.99 (0.304) | 0.000 | 0.000 |
| PEMS07 | SimpleTM | 38.00 (0.139) | 23.36 (0.085) | - | - |
| | TimeMixer | 40.65 (0.498) | 25.44 (0.363) | 0.000 | 0.000 |
| PEMS08 | SimpleTM | 27.42 (0.114) | 17.09 (0.069) | - | - |
| | TimeMixer | 32.58 (2.453) | 20.68 (1.776) | 0.000 | 0.000 |

without any increase in model parameters. The pooled standard deviations, as shown in Table 11, are small across all datasets, indicating that the performance advantages are stable/reproducible. To test statistical significance, we performed a Type II ANOVA analysis with the null hypothesis that *there is no difference in performance between baseline model with and without geometric attention.* The consistently low p-values ($p < 0.05$) across all datasets confirm that the observed improvements are statistically significant and not attributable to random chance or prediction length variability.

**Stationary wavelet transform.** For the SWT, we conducted three types of ablation experiments: (i) complete removal of the SWT decomposition and reconstruction; (ii) replacement with the Fast Fourier Transform (FFT) as the tokenizer and inverse FFT as the de-tokenizer; (iii) replacement with parameter-matched 1-D convolution layers to ensure fair comparison. The Performance/Parameters $\Delta\%$ column in Table 11 shows the percentage change in performance and total trainable parameters relative to the baseline model.

The results in Table 11 suggest that removing SWT leads to substantial performance degradation, particularly evident in the Solar dataset where we observed a $34.8\%$ MSE increase despite only

reducing parameters by 5.27%. Even parameter-matched alternatives underperformed compared to our model: replacing SWT with equivalent convolutions increased MSE by 16.8% on Solar and 9.8% on Weather. Additionally, FFT-based variants showed performance drops, with 20.7% and 9.8% MSE increases on Solar and Weather, respectively. All of these differences are statistically significant ($p < 0.01$), showing that both architectural components contribute to the performance of our simple baseline presented here. Particularly interesting is the substantial performance gap between SWT and its convolution-based replacement, which suggests that SWT's effectiveness stems from its multi-resolution analysis capabilities rather than merely adding model capacity.

Table 11: Ablation study results for different models across various datasets. Metrics include Mean Squared Error (MSE) and Mean Absolute Error (MAE) with their pooled standard deviations (SD), along with percentage changes in performance and parameter counts relative to the baseline model. Lower MSE/MAE values indicate better performance, while a negative performance delta signifies performance degradation.

| Dataset | Model | Number of parameters changes? | MSE (Pooled SD) | Performance worse? | MAE (Pooled SD) | Performance worse? | MSE p-value | MAE p-value |
|---|---|---|---|---|---|---|---|---|
| ETTh1 | SimpleTM | - | 0.422 (0.0015) | - | 0.428 (0.0007) | - | - | - |
| | w/o GeomAttn | None | 0.437 (0.0010) | Yes, by -3.55% | 0.440 (0.0009) | Yes, by -2.80% | 0.000 | 0.000 |
| | w/o SWT | Yes, by -0.436% | 0.432 (0.0050) | Yes, by -2.37% | 0.435 (0.0043) | Yes, by -1.60% | 0.000 | 0.000 |
| | Conv-SWT | None | 0.433 (0.0063) | Yes, by -2.60% | 0.435 (0.0035) | Yes, by -1.60% | 0.000 | 0.000 |
| | FFT-SWT | Yes, by -0.436% | 0.433 (0.0063) | Yes, by -2.60% | 0.435 (0.0040) | Yes, by -1.60% | 0.000 | 0.000 |
| ETTm1 | SimpleTM | - | 0.381 (0.0009) | - | 0.396 (0.0008) | - | - | - |
| | w/o GeomAttn | None | 0.385 (0.0011) | Yes, by -1.05% | 0.398 (0.0009) | Yes, by -0.51% | 0.000 | 0.000 |
| | w/o SWT | Yes, by -0.436% | 0.386 (0.0031) | Yes, by -1.31% | 0.398 (0.0021) | Yes, by -0.51% | 0.000 | 0.002 |
| | Conv-SWT | None | 0.389 (0.0025) | Yes, by -2.10% | 0.399 (0.0020) | Yes, by -0.76% | 0.000 | 0.000 |
| | FFT-SWT | Yes, by -0.436% | 0.390 (0.0017) | Yes, by -2.36% | 0.399 (0.0009) | Yes, by -0.76% | 0.000 | 0.000 |
| Solar | SimpleTM | - | 0.184 (0.0016) | - | 0.247 (0.0031) | - | - | - |
| | w/o GeomAttn | None | 0.194 (0.0123) | Yes, by -5.43% | 0.253 (0.0127) | Yes, by -2.40% | 0.018 | 0.120 |
| | w/o SWT | Yes, by -5.27% | 0.246 (0.0010) | Yes, by -34.8% | 0.289 (0.0008) | Yes, by -17.0% | 0.000 | 0.000 |
| | Conv-SWT | None | 0.215 (0.0125) | Yes, by -16.8% | 0.273 (0.0158) | Yes, by -10.5% | 0.000 | 0.000 |
| | FFT-SWT | Yes, by -5.27% | 0.222 (0.0187) | Yes, by -20.7% | 0.284 (0.0198) | Yes, by -15.0% | 0.000 | 0.000 |
| Weather | SimpleTM | - | 0.243 (0.0005) | - | 0.271 (0.0007) | - | - | - |
| | w/o GeomAttn | None | 0.245 (0.0021) | Yes, by -0.82% | 0.273 (0.0018) | Yes, by -0.74% | 0.007 | 0.032 |
| | w/o SWT | Yes, by -0.426% | 0.247 (0.0014) | Yes, by -1.65% | 0.274 (0.0006) | Yes, by -1.11% | 0.000 | 0.000 |
| | Conv-SWT | None | 0.267 (0.0006) | Yes, by -9.88% | 0.285 (0.0005) | Yes, by -5.17% | 0.000 | 0.000 |
| | FFT-SWT | Yes, by -0.426% | 0.267 (0.0008) | Yes, by -9.88% | 0.286 (0.0005) | Yes, by -5.54% | 0.000 | 0.000 |

## C.2 Additional Ablations

**Inter-channel dependencies.** In our tested datasets, recent results (e.g., PatchTST Nie et al. (2023)) have shown that individual channels were often sufficient for making reasonable forecasts, indicating limited direct correlations between channels. However, our experiments show that incorporating all channels in the token embedding improves forecasting performance compared to single-channel embeddings (where the bivector reduces to a scalar), as shown in Table 12.

This improvement likely comes from how our attention mechanism uses the channel information. While not explicitly mixing channels through projection layers, it computes attention weights using all channels simultaneously as well as using cross-channel relationship through the wedge product. This allows features across all channels to collectively determine how much each token's full channel vector contributes to the final representation, creating an implicit form of channel interaction. We hypothesize that the model adaptively captures useful channel relationships when they exist, while avoiding imposing artificial correlations when they do not.

Table 12: Performance comparison of SimpleTM with and without the Independence feature across three datasets. Metrics include Mean Squared Error (MSE) and Mean Absolute Error (MAE) with pooled standard deviations (SD), along with p-values for statistical significance.

| Dataset | Model | MSE (Pooled SD) | MAE (Pooled SD) | MSE $p$-value | MAE $p$-value |
|---|---|---|---|---|---|
| ETTh1 | SimpleTM | 0.422 (0.0015) | 0.428 (0.0007) | - | - |
| | + Independence | 0.451 (0.0073) | 0.444 (0.0035) | 0.000 | 0.000 |
| ETTm1 | SimpleTM | 0.381 (0.0009) | 0.396 (0.0008) | - | - |
| | + Independence | 0.394 (0.0079) | 0.400 (0.0055) | 0.000 | 0.007 |
| Weather | SimpleTM | 0.243 (0.0005) | 0.271 (0.0007) | - | - |
| | + Independence | 0.268 (0.0024) | 0.286 (0.0019) | 0.000 | 0.000 |

## D    EFFICIENCY ANALYSIS

To provide a thorough efficiency comparison, we evaluated our model against two of the most competitive baselines: the transformer-based iTransformer Liu et al. (2024a) and linear-based TimeMixer Wang et al. (2024). Our experimental setup used a consistent batch size of 256 across all models and measured four key metrics: total trainable parameters, inference time, GPU memory footprint, and peak memory usage during the backward pass. Results for all baseline models were compiled using PyTorch.

Our findings demonstrate remarkable efficiency improvements: On the Weather dataset, our model achieves better accuracy while using only 0.3% of iTransformer's parameters (13K vs 4.8M) and 13% of TimeMixer's parameters (13K vs 104K). Our memory footprint is 38% smaller than iTransformer's and 66% smaller than TimeMixer's. In terms of speed, our model is 1.7x faster than iTransformer and 3.4x faster than TimeMixer. These efficiency gains are even more pronounced on the larger Solar-Energy dataset, where our model uses just 1.3% of TimeMixer's parameters (166K vs 13M) while achieving 24% better accuracy. Our memory consumption is 73% lower than TimeMixer's, and inference speed is 5.8x faster. Notably, these improvements come without compromising performance, as our model maintains superior or comparable MSE scores across both datasets.

In the reported experiments, we prioritize memory and computation efficiency by choosing the bivector's magnitude for the reduction function $\zeta(\cdot)$. However, we have a fair bit of flexibility to upgrade the reduction function later for additional performance gains.

Table 13: Comparison of model performance and resource utilization across different datasets. Metrics include Mean Squared Error (MSE), total parameter count, inference time (seconds), GPU memory footprint (MB), and peak memory usage (MB).

| Dataset | Model | MSE | Total Params | Inference Time (s) | GPU Mem Footprint (MB) | Peak Mem (MB) |
|---------|-------|-----|--------------|--------------------|------------------------|---------------|
| | SimpleTM | 0.162 | 13,472 | 0.0132 | 994 | 181.75 |
| Weather | TimeMixer | 0.164 | 104,433 | 0.0453 | 2,954 | 2,281.38 |
| | iTransformer | 0.176 | 4,833,888 | 0.0222 | 1,596 | 847.62 |
| | SimpleTM | 0.163 | 166,304 | 0.0455 | 2,048 | 1,181.56 |
| Solar | TimeMixer | 0.215 | 13,009,079 | 0.2644 | 7,576 | 6,632.40 |
| | iTransformer | 0.203 | 3,255,904 | 0.0663 | 4,022 | 2,776.50 |

**Potential benefits from Parallelization.** While our current implementation follows a standard SWT approach, additional opportunities for further optimization are possible. In the standard SWT, each level's approximation is convolved with an upsampled wavelet filter to yield the next level's approximation and detail coefficients. Because no downsampling occurs, there is no inherent data dependency between levels, yet an iterative formulation does not exploit this property. By recognizing that each scale can be expressed directly as a convolution of the original signal with an appropriately composite filter, one can bypass the step-by-step procedure and compute all levels in parallel. This approach leverages convolution's associativity to collapse the iterative chain into a set of direct convolutions, each mapping from the original input to a given scale's approximation or detail.

Let $h_{\uparrow 2^{s-1}}$ and $g_{\uparrow 2^{s-1}}$ denote the upsampled low-pass and high-pass filters at level $s$ by inserting $2^{s-1}$ zeros between each tap. We can define composite filters $H_s$ and $G_s$ via the recurrences

$$H_s = H_{s-1} * h_{\uparrow 2^{s-1}} \quad \text{and} \quad G_s = H_{s-1} * g_{\uparrow 2^{s-1}}, \text{ for } s > 1,$$

starting from $H_1 = h$ and $G_1 = g$. This recurrence builds up the composite filter by successively convolving the upsampled filters from previous levels, capturing the cumulative filtering. By associativity, the outputs at level $s$ become

$$a^{(s)} = u * H_s \quad \text{and} \quad d^{(s)} = u * G_s,$$

thus eliminating the need to explicitly compute $a^{(s-1)}$ first. This parallelized approach would allow all convolutions to be applied directly to the original signal $u$ and launched simultaneously on modern hardware accelerators, potentially reducing computational overhead in real-time or large-scale

wavelet-based applications. The direct implementation of these convolutions using the composite filters would involve operations with dense matrices, giving up at least some of the parallelization gains. To address this, further optimization can be achieved by approximating these convolutions using structured matrices, such as circulant matrices. This circulant approximation replaces the upsampled filters with wrapped versions, allowing the use of FFT. Here, we are leveraging the shift-invariance of circulant matrices, with the understanding that trade-offs exist between speed and accuracy. We leave the implementation and evaluation of these modifications for future work, but note that it is a promising direction for improvements.

# E   SHOWCASING AND CHECKING FORECASTING CAPABILITIES

Our forecasting model demonstrates its ability to predict trends across various time series datasets, including ECL, Traffic, Solar Energy, and Weather. Each example uses a 96-step input to generate 96-step predictions. In the visualizations, the blue lines represent the lookback window, the orange lines indicate the ground truth forecasting window, and the red lines show the model's predictions. The model's strengths lie in pattern recognition and trend prediction. It is good at identifying and extrapolating recurring patterns, particularly evident in the Traffic dataset (Fig 8), where it accurately captures cyclical nature and oscillations. In the Solar-Energy dataset (Fig 10), the model successfully predicts overall directional trends.

However, there are areas for improvement. The model sometimes struggles with precise amplitude prediction, as seen in the ECL dataset, where predicted peaks and troughs do not always align perfectly with the ground truth. Phase shifts between predicted and actual values are also observed in some Traffic dataset forecasts (Fig 9), suggesting a need for improved timing mechanisms. Handling anomalies shows another challenge. The model occasionally struggles with sudden spikes or dips, particularly evident in the Solar Energy dataset. Additionally, in longer predictions, the model shows signs of instability or drift, as observed in certain forecasts for the ECL and Weather dataset.

In summary, while the model demonstrates adaptability to different scales and patterns, there's room for improvement in amplitude accuracy, phase alignment, anomaly handling, and long-term stability. Future work should focus on addressing these limitations to enhance the model's robustness and accuracy across diverse time series forecasting tasks.

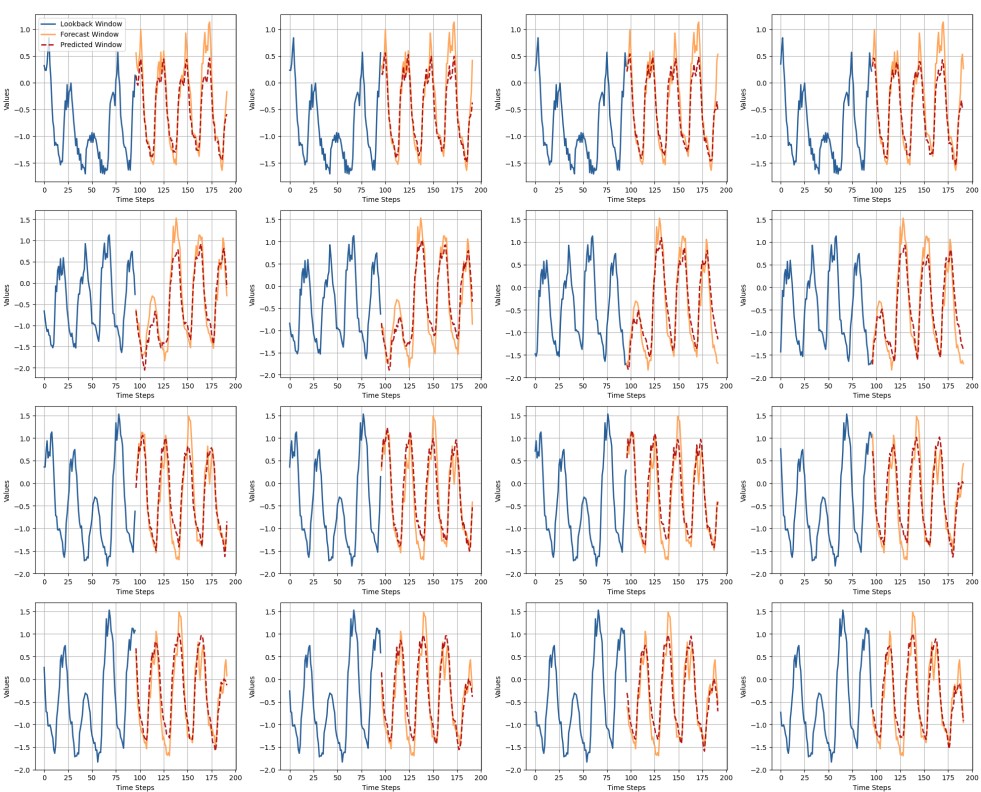

Figure 8: Forecasting examples from the ECL dataset with a 96-step input and 96-step predictions.

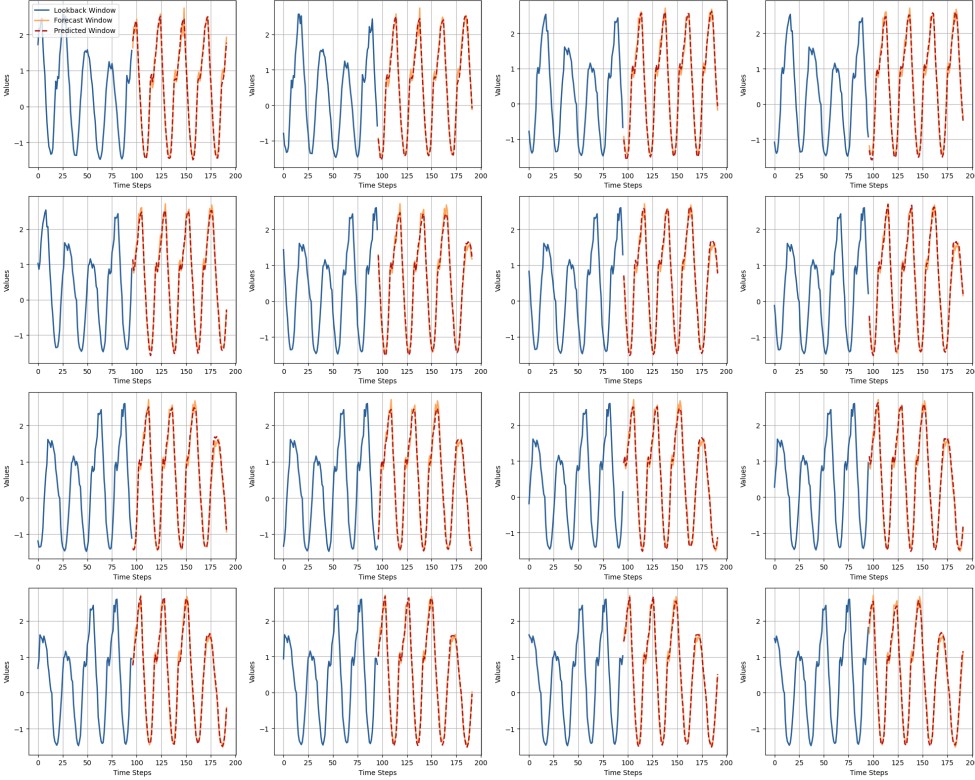

Figure 9: Forecasting examples from the Traffic dataset with a 96-step input and 96-step predictions.

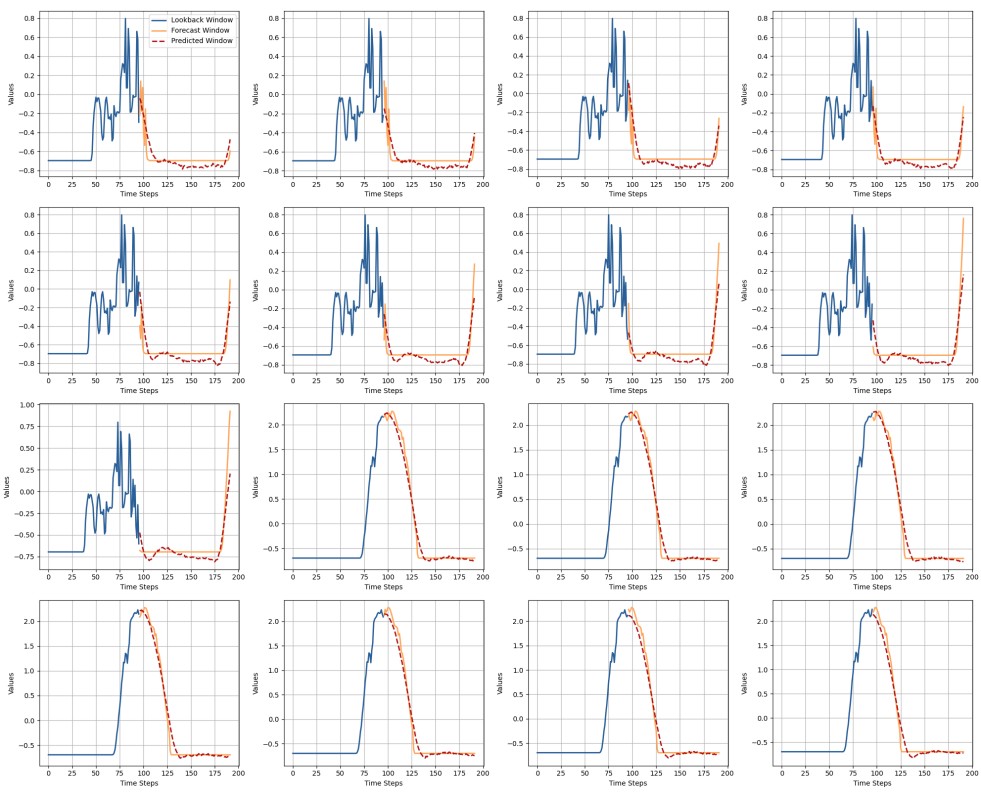

Figure 10: Forecasting examples from the Solar dataset with a 96-step input and 96-step predictions.

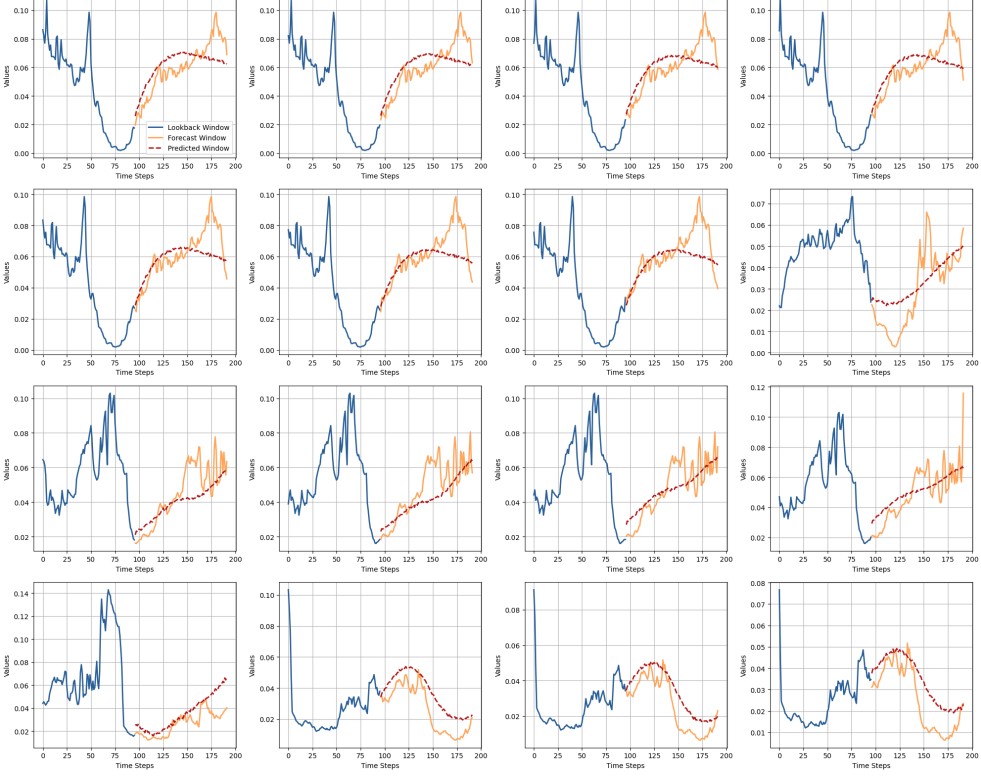

Figure 11: Forecasting examples from Weather dataset with a 96-step input and 96-step predictions.

