# OpenReview forum: "SimpleTM: A Simple Baseline for Multivariate Time Series Forecasting"
_ICLR.cc/2025/Conference — ICLR 2025 Poster_

### Official Review · Reviewer_8zFK · 2024-10-31

**Soundness:** 3
**Presentation:** 3
**Contribution:** 3
**Rating:** 8
**Confidence:** 3

**Summary:**

The authors present a simplified approach to multivariate time series forecasting by using stationary wavelet transforms (SWT) for tokenization and geometric product attention. The use of geometric product attention allows the model to capture both the magnitude-based similarity and geometric relationships between tokens. The authors state that even a single or two layer model, using this architecture, yields results competitive with much bigger models on most benchmarks reported in the literature.

**Strengths:**

Originality and Significance: The paper leverages SWTs for time series tokenization, a concept inspired by recent successes in image processing with transformers. The authors go beyond simply applying SWTs by introducing learnable filters within the SWT framework. Further, the paper's most original contribution lies in its introduction of the geometric product attention mechanism. All of these contribute to the originality of the paper.

Quality: The paper demonstrates the effectiveness of its approach through rigorous experimental evaluation on a diverse range of benchmark datasets. The model achieves state-of-the-art performance on numerous long-term and short-term forecasting tasks, outperforming established baselines like TimeMixer and iTransformer.

**Weaknesses:**

Clarity: Sections explaining the method (3,4,5) are too long, and can be summarized first before delving into the details. Also, the contributions of the paper needs to be summarized in the introduction, which is not currently done. (Note that I currently don't consider these points in my score, and they are suggestions)

More empirical evaluation necessary. These are the major points that made me choose a score of 6 for this paper. Addressing them can potentially increase my score:

* It would be worth adding simple baselines like ARIMA and ETS, just so we know how much better the transformer-based models are.
* At the same time, it would be worth adding a note on the number of parameters in each model, and the inference time of each model, which are both important aspects for practical forecasting.
* Especially since a new geometric product attention mechanism is proposed, detailing the memory footprint and computational complexity of the mechanism compared to that of other models would be useful.
* The results don't report the standard error. It's not unclear how many seeds were used in the experiments for each model.
* The authors claim that the mechanism captures inter-channel dependencies, but there is no empirical evidence that this is being useful for forecasting. An ablation to demonstrate that would be useful. You can plot the interchannel dependencies learned for each dataset and also show that meaningful dependencies were learnt.
* Details on training: Since the baselines and the proposed model were trained for each dataset, the authors should provide details on how the baselines were trained (how hyperparameter selection was done, what protocol was used etc.) in the appendix. It is important to check if the evaluation compares all models in a fair manner. This is important to check how sensitive the model is to different hyperparameter values.

**Update**: My concerns are addressed. Thanks to the authors for the efforts. I increase my score to 8, but I want to note that I consider this paper a 7, considering the strengths of the paper and the additions that the authors have done (I am unable to choose 7 in OpenReview).

**Questions:**

Please see weaknesses.

---

### Official Review · Reviewer_avrv · 2024-11-02

**Soundness:** 2
**Presentation:** 3
**Contribution:** 2
**Rating:** 6
**Confidence:** 4

**Summary:**

This paper introduces a simplified transformer-based model for multivariate time series (MTS) forecasting that performs competitively with larger models while using only 1-2 layers. The proposed architecture includes three key innovations:

Stationary Wavelet Transform (SWT) Tokenization: The authors apply SWT to efficiently capture local and global dependencies across different time frequencies. Unlike discrete wavelet transforms, SWT maintains time invariance by avoiding downsampling, allowing the model to capture both short- and long-term patterns with high temporal resolution.

Geometric Product Attention Mechanism: To address the multidimensionality in MTS data (including variate, time, and frequency dependencies), the authors adapt a Clifford Algebra-inspired attention mechanism to capture both magnitude-based and spatial orientation relationships among tokens. This lightweight modification enhances interpretability and accuracy compared to the traditional dot-product self-attention.

Learnable Inverse SWT (ISWT) for Reconstruction: Finally, the model employs a learnable ISWT module to reconstruct the MTS from the processed time-frequency tokens, facilitating a more accurate and coherent forecast by synthesizing the multi-scale information captured in the initial stages.

Experiments and Analysis:
The authors evaluate their model on diverse MTS datasets for both short- and long-term forecasting, demonstrating competitive or superior performance relative to established baselines. They also perform ablations, analyze learned filter effects in decomposition, and examine multi-scale forecasts, showcasing the model’s ability to capture both macro trends and finer details across scales.

**Strengths:**

- While wavelet transforms have been applied in forecasting before, this is the first known application for tokenization within a transformer architecture for time series. The approach is well-motivated, addressing the limitations of viewing time series as a single token stream and effectively handling various seasonalities and frequencies. The proposed end-to-end architecture builds on this foundation and includes complementary modules to support the overall design.

- The paper is largely well-written, with comprehensive mathematical explanations for each module. Visuals effectively clarify the concepts, making the technical details accessible.

**Weaknesses:**

- Results in Tables 1 and 2 lack statistical tests to validate significance. For instance, in Table 1, differences between the proposed model and the next-best model are minimal for half of the datasets, with changes ranging from 0.001 to 0.004, which may not be statistically significant.

- The ablation study is challenging to interpret. While the authors mention component replacement and removal experiments, the table only compares geometric versus vanilla attention across datasets. It’s unclear whether the study assesses the contribution of the Stationary Wavelet Transform (SWT), which is fundamental to the architecture.

- The authors claim their model is competitive with much larger, even LLM-based, models. However, there’s no discussion of model sizes of each baselines. Moreover there is no LLM-based baselines in the comparisons (correct me if I am wrong), which weakens this claim.

- The TimeMixer results in Table 1 is not directly comparable to results shared in their official repository. With the official TimeMixer results, the proposed model’s competitive standing may be less compelling, though further exploration could be valuable if the proposed model size is indeed significantly smaller as authors claim.

Things to improve the paper that did not impact the score:

- The introduction reads more like a related work section, making it difficult to grasp the proposed method and its motivation. It heavily references prior work, which distracts from understanding the main contribution. Much of this content might be more suitable for the related work section.

- A background section on Wavelet transformations would be helpful for readers unfamiliar with the concept. Currently, such readers may need to consult external sources to follow the paper. If space is limited, this background could be included in the appendix.

- The illustrations in Figures 6 and 7 are quite small, which makes them difficult to interpret. Enlarging these figures would improve readability.

**Questions:**

Overall, this paper has the potential to be a good contribution because the motivation is sound. After getting clarifications for the concerns below, mostly on the experiments and their results, I would consider increasing my score.

- For the lack of statistical significance tests: Could the authors provide statistical tests for Tables 1 and 2 to demonstrate whether the observed performance differences are statistically significant?

- Regarding the ablation study: Could the authors clarify if the ablation study evaluates the impact of the Stationary Wavelet Transform (SWT) module, and if so, how this contribution is isolated and measured?

- For LLM-based models: Is any of the baselines LLM-based? If not, how does thus the claim that the proposed model is competitive with much larger, even LLM-based models, verified?

- For model size comparison: Could the authors include model size comparisons and discuss how their model compares in size and parameter count to the baselines? Moreover the discussion on compute and memory footprint can be extended to multiple competitive baselines to get a better understanding of where the proposed model stands.

- About TimeMixer results: Could the authors confirm whether the TimeMixer results were obtained from scratch or directly from prior work? Have they checked the results from the official TimeMixer repository (https://github.com/kwuking/TimeMixer)? Both hyperparameter-tuned and non-tuned results seem much better than the ones reported in the paper.

---

### Official Review · Reviewer_Lbi6 · 2024-11-03

**Soundness:** 3
**Presentation:** 3
**Contribution:** 2
**Rating:** 5
**Confidence:** 3

**Summary:**

The paper presents a new model to do multivariate point forecasting.

The model is based on a transformer architecture, with two main modifications. The first one is that the usual dot product in the attention mechanism is replaced by a combination of a dot product and a wedge product. The second one is that the original input, after being transformed by a trained linear layer, is then transformed by a trained wavelet layer into various channels which contains information about various time frequencies.

The model is then compared with various state-of-the-art point forecasting models, and comes up ahead. An annealing is also made where the model is tested against itself, but without its wedge product modification of the attention mechanism, and it shows that the version with the wedge product is slightly better than the version without it.

**Strengths:**

1. The model is compared with a vast array of recent timeseries models, on standard testing datasets.
2. The idea of using a wedge product in the attention mechanism to allow the model to give more weight to dissimilar tokens seems novel.

**Weaknesses:**

1. A lot of important hyperparameters of the models are missing. Such as $\zeta$, $L'$, and others.
2. The training procedure is missing. It would be useful to know how reliable the results are when training the models multiple times from various randomized initial parameters.
3. Ablation experiments are lacking. Firstly, the minimal score improvement when adding the wedge product to the attention mechanism could easily be caused by the associated addition of more trainable parameter. A proper comparison should be done where the extra parameters are added to the traditional scalar product version of the attention mechanism. Secondly, the usefulness of the wavelet transform should also be tested through ablation. For example, by having replacing by two linear layer (or a MLP), one in the encoding, and one in the decoding part of the model.
4. More detailed information should be provided concerning the number of parameters and memory footprint of this model, in particular in how they compare to those of other models. One concerning point is that only a constant memory footprint is provided, while the number of tokens sent to the attention layer should grows with the dimensionality of the dataset, which should therefore lead to a variable memory footprint.

**Questions:**

1. In figure 6b, the filtering procedure makes the correlations between time steps very close to 1, and never negative. This is counterintuitive, since there doesn't seem to be any bias which should favor positive correlations instead of negative correlations in the wavelets parametrization. Also, having only correlations close to 1 should make it harder for a model to distinguish the information contained in the initial signal. Do you have an explanation for how the model is still able to function with this correlation pattern?
2. Some datasets have a tendency to have almost no meaningful correlation between channels (the single channel forecast being mostly enough to do good forecasts). Do you know whether the datasets you have tested your model have this property or not?

---

### Official Review · Reviewer_L7y3 · 2024-11-06

**Soundness:** 3
**Presentation:** 3
**Contribution:** 3
**Rating:** 8
**Confidence:** 3

**Summary:**

The authors propose a model for multivariate time-series (MTS) forecasting that utilizes a stationary wavelet transform (SWT) for tokenization, a geometric product-based self-attention mechanism to capture inter-channel relationships, and a learnable inverse SWT to reconstruct the signal in the time domain. The experiments show that the model demonstrates competitive performance on both long-term and short-term forecasting tasks, outperforming state-of-the-art methods like TimeMixer and iTransformer on a variety of datasets. The ablation experiments show that the geometric attention mechanism consistently improves performance, and that the wavelet-based tokenization provides an inductive bias for learning data-driven filters. Despite the above, the authors note that while the model is effective and lightweight, it may not scale well to larger datasets or more complex tasks, such as token-by-token generation. They also emphasize the potential for their model components to inform future work in specialized tokenization and embedding schemes.

**Strengths:**

* The paper is well written with easily understandable structure

* The experiments are extensive, covering 8 multi-variate time series datasets and 15 baseline models that were compared against the proposed model

* The proposed tokenization and attention schemes show notable performance improvements over previous works across datasets. The idea is simple and effective as demonstrated by the experiments. It is worth noting that the model is computationally efficient with a low memory footprint, making it feasible for resource-constrained settings

**Weaknesses:**

* While the model performs well on smaller datasets, the authors acknowledge that it may not scale efficiently to larger datasets or more complex tasks. The model's architecture is not well-suited for token-by-token generation, and larger models may be needed as datasets grow in size.

* The paper could be strengthened by including details on the number of parameters in the proposed model compared to other large-scale models, as well as results exploring the impact of increasing the number of layers in the proposed model. Providing additional information about the characteristics of the datasets used and the hyperparameter search settings would also improve clarity and improve reproducibility

**Questions:**

* Appendix is referenced in Table 1 but does not exist in the submission

* The authors mention that the difference in results between fixed and learned wavelet filters is minimal - is there insight into why this is the case? In addition methods such as discrete fourier transforms are widely used in time series analysis. While the use of wavelet transform is well motivated, it would be beneficial to characterize other methods in context of the proposed tokenization scheme

* Experimental results for TimeMixer, the closest baseline to the proposed model, differ from those reported in the original paper, for both long and short term forecasting. Where does the difference come from? Error bars will help to assess if differences might be significant or not

---

> ### Comment · Reviewer_L7y3 · 2024-11-27
>
> Thank you to the authors to address all my questions and concerns. Based on their response, I'm happy to revise (upgrade) my score. I feel with the appropriate revisions it's a good paper.

---

> > ### Author Response · Authors · 2024-11-27
> >
> > Dear Reviewer L7y3,
> >
> > Thank you for your positive feedback and for upgrading your score. We are glad that we have addressed your concerns. We are grateful for your support throughout the review process.
> >
> > Best regards,
> >
> > The Authors.

---

### Meta-Review · Area_Chair_fBdt · 2024-12-21

**Metareview:**

This paper introduces a lightweight model for multivariate forecasting that utilizes a stationary wavelet transform (SWT) for tokenization, and a geometric product-based self-attention mechanism. Reviewers  appreciated the simple, efficient and well-motivated architectural choices and the strong performance of the model, both on quality and computational resources. The paper is well written and the experiments and ablation studies are  comprehensive and thorough.

 The AC is glad to recommend acceptance of the paper. Please address the reviewer comments and  the new experimental results in the camera-ready version. Also, please consider reporting experimental results using longer context lengths (this will enable a fairer comparison against large zero-shot models, and baselines like PatchTST which are known to be much more competitive on longer context lengths)

**Additional Comments On Reviewer Discussion:**

The reviewers appreciated the simplicity and effectiveness of the model, but asked for more ablation studies, memory/inference time comparisons, more baselines. and details of model training and hyperparameters. The authors added ablation studies related to tokenization, wedge attention and channel independence, and also added more baselines such as Arima and ETS. They also added memory and inference-time benchmarks, added statistical significance tests, and included more details of their hyper-parameters and dataset characteristics.

---

### Decision · Program_Chairs · 2025-01-22

Accept (Poster)